# Multichannel optogenetics combined with laminar recordings for ultra-controlled neuronal interrogation

David Eriksson [1,4✉], Artur Schneider[1,6], Anupriya Thirumalai [1,5,6], Mansour Alyahyay[1], Brice de la Crompe[1], Kirti Sharma [2], Patrick Ruther [2] & Ilka Diester [1,3✉]

Simultaneous large-scale recordings and optogenetic interventions may hold the key to deciphering the fast-paced and multifaceted dialogue between neurons that sustains brain function. Here we have taken advantage of thin, cell-sized, optical fibers for minimally invasive optogenetics and flexible implantations. We describe a simple procedure for making those fibers side-emitting with a Lambertian emission distribution. Here we combined those fibers with silicon probes to achieve high-quality recordings and ultrafast multichannel optogenetic inhibition. Furthermore, we developed a multi-channel optical commutator and general-purpose patch-cord for flexible experiments. We demonstrate that our framework allows to conduct simultaneous laminar recordings and multifiber stimulations, 3D optogenetic stimulation, connectivity inference, and behavioral quantification in freely moving animals. Our framework paves the way for large-scale photo tagging and controlled interrogation of rapid neuronal communication in any combination of brain areas.

[1] Optophysiology IMBIT//BrainLinks-BrainTools Faculty of Biology, Institute III, University of Freiburg, Georges-Köhler-Allee 201, 79110 Freiburg i. Br., Germany. [2] Department of Microsystems Engineering (IMTEK). BrainLinks-BrainTools, University of Freiburg, Georges-Köhler-Allee 102, 79110 Freiburg i. Br., Germany. [3] Bernstein Center Freiburg. Hansastr. 9a, 79104 Freiburg i. Br., Germany. [4] Present address: Institute of Physiology, University of Freiburg, Hermann-Herder-Str. 7, 79104 Freiburg, Germany. [5] Present address: Institute for Auditory Neuroscience and InnerEarLab, University Medical Center Göttingen, 37075 Göttingen, Germany. [6] These authors contributed equally: Artur Schneider, Anupriya Thirumalai.
✉email: david.eriksson@physiologie.uni-freiburg.de; ilka.diester@biologie.uni-freiburg.de

Combined optogenetic manipulations and electrophysiological recordings have been pioneered with a two-tipped optrode[1] and optetrodes, i.e., an optical fiber surrounded by tetrodes[2]. This has been refined with a single-tip approach[3], integrated fibers in electrode arrays[4], and the use of micro light-emitting diodes (μ-LEDs) combined with recording probes or co-integrated in silicon-based probes[5–8]. Fiber-based approaches allow the flexible targeting of multiple small and large areas while maintaining the flexibility to apply any desired wavelength via an external exchangeable light source. The existing fiber-based approaches illuminate the tissue surrounding the electrode, either from a fiber at the top of the electrode shank[9–11] or from a tapered fiber at some distance in front of the electrode shank[12]. Although tapered fibers produce more even illumination along the electrode shank, the light is typically strongest toward the fiber tip[13,14]. Furthermore, the relatively stiff large-diameter fibers render the fiber location in the tissue dependent on the location of the connector[15], and for multiple fibers, it may become difficult to avoid rupturing the vasculature. Moreover, multifiber stimulation would require a multichannel optical commutator for long-term stimulation in a freely moving animal. Finally, it is a commonly encountered problem that optical stimulation next to extracellular electrodes causes photoelectrochemical and electromagnetic interference and photovoltaic effects[16,17], not to mention the mechanical damage induced, especially by the larger fibers[18,19], which diminishes the recording quality.

In this work, we propose an alternative optrode design for silicon probes based on multiple ultrathin optical fibers, which we call Fused Fiber Light Emission and eXtracellular Recordings (FFLEXR). We mitigate the above-mentioned issues with thin, linearly emitting optical fibers that can be attached to any silicon probe, depth-resolved stimulation, a lightweight fiber-matrix connector, a flexible multifiber ribbon cable, an optical commutator for efficient multichannel stimulation, a general-purpose patch cord, and a photovoltaic response (PVR) management algorithm.

## Results

**Multifiber commutation in the freely moving animal**. To allow simultaneous extracellular recordings and multichannel fiber targeting in a freely moving animal, we developed an optical commutator that is integrated into an electrical commutator (Fig. 1A). Light from a laser is guided to one of four patch cord fibers via a galvo scanner and a scan lens. The fibers' locations are determined by scanning the optical ferrule, which channels the light into the patch cord, and by measuring the transmitted light with a linear intensity meter clip attached to the animal patch cord (Fig. 1A and SFig. 1A, B). Optical fibers separated with a distance of 100 μm can be addressed individually with crosstalk of 0.3% (SFig. 1C), leading to a maximum of 100 individually addressable fibers per square millimeter in the ferrule. We were able to couple 49.3 ± 7.3% of the input power into a 60 μm fiber with a 50 μm core and 14.3 ± 6.6% of the input power into a 30 μm fiber with a 24 μm core (SFig. 1D). An angular encoder allowed the galvo scanner to track one of the selected fibers as the animal, motorized commutator, and ferrule rotated. The average mechanical error of the optical commutator was 3.3 μm (SFig. 1E).

To achieve high-quality extracellular recordings along with large-scale optical interventions, we implanted optical fibers with outer and core diameters of 30 and 24 μm, respectively (Fig. 1B). In comparison to fibers with a larger outer diameter, the thin 30 μm fiber was very flexible (Fig. 1C), allowing the implantation location to be independent of the fiber connector location

(Fig. 1D). The fibers in the patch cord had a core diameter of 50 μm, and each such fiber targeted two or four 30 μm fibers in the animal (Fig. 1E, F).

The fiber-matrix connector was manufactured from silicon using photolithography and deep reactive-ion etching (DRIE) (Fig. 1G, H and SFig. 2A, B). Using 1 mm-diameter precision guide pins (SFig. 2C) with a 0.995 mm-diameter lower boundary, the alignment of two-fiber connection plates was 5.2 ± 1.7 μm for the 30 μm fiber and 9.4 ± 0.7 μm for the 65 μm fiber (Fig. 1I), and the light transmission efficiency was on the order of 50 ± 11% for the 30 μm fiber and 45 ± 15% for the 65 μm fiber, with minimal crosstalk between neighboring fibers (Fig. 1J).

**Thin sidelight fibers for Lambertian emission**. We manufactured linearly emitting fibers from 30 μm fibers by manually polishing them with rotating 5 μm diamond paper (SFig. 3A). The result was a linearly emitting fiber with an axial component (rotationally symmetric and forward-directed) stronger than the radial component (rotationally symmetric and sideward-directed) (SFig. 3Bi, 3Bii, and C). To approach Lambertian emission (with a radial component stronger than the forward component), we coated the fiber with a thin diffusing layer (Fig. 1K and SFig. 3Biii, 3Biv, and D). In addition to the diffusive property, this layer has a refractive index of ~1.4[20], which is closer to the refractive index of the fiber core (~1.5) than the refractive index of water (~1.33). Therefore, the diffusive coating facilitates the escape of light from the fiber. A broad range of emission lengths (0.5–10 mm) can be produced using this simple procedure (SFig. 3E–G). The ability to control how much light is emitted at each position along with the fiber independently (through manual polishing), together with the diffusing layer, results in more even emission along with the fiber relative to the Lambda probe (SFig. 3H–J).

**Back emission fibers for laminar probes**. To manufacture the implant, two coated or uncoated side-emitting fibers were attached to the back (i.e., facing away from the electrode contacts) of a 75 μm-wide silicon probe (SFig. 4A). To test the validity of this back emission (BE) probe, it was surrounded by four additional fibers (hereinafter referred to as the "fiber matrix") at a distance of 700 μm (SFig. 4B). Light with a wavelength of 596 nm decays to 50% at 800 μm along the main emission axis[10], which in our case is radially from the sidelight fiber. If the response from the back emission probe is on par with that of the surrounding fibers, it means that it is feasible to put fibers behind laminar probes to stimulate brain tissue. The same configuration was used for the other hemisphere as well. The BE probes, fiber matrix, light connector, and two electrode connectors were temporarily supported by a three-dimensional (3D) printer holder during implantation (SFig. 4C). The total weight of the connector with an epoxy-filled 3D-printed connector sleeve (SFig. 4C) was 0.25 g. Implantation was guided by in vivo fluorescence (SFig. 4D). The 4 mm-long fibers did not bend during implantation, and dimpling of the brain was minimal when the BE probes and fibers were implanted (SMovie 1).

To demonstrate the viability of our approach in freely moving animals, we implanted seven rats with combinations of the BE probe and the fiber matrix (see Tables 1 and 2). We recorded electrophysiological signals and applied a 589 nm laser source (SFig. 5A) in freely moving rats in sessions of 60–120 min over the course of 10 days. To test the light distribution along with the optical fiber, we took advantage of the PVR amplitude (SFig. 5B) along the electrode shank (SFig. 5C). For the uncoated side-emitting fiber, there was a significantly larger PVR toward the tip of the electrode ($p = 0.035$, $n = 4$, Student's $t$-test, slope = $-0.96 \pm 0.52$ mm$^{-1}$), while for the coated fiber, the difference was not

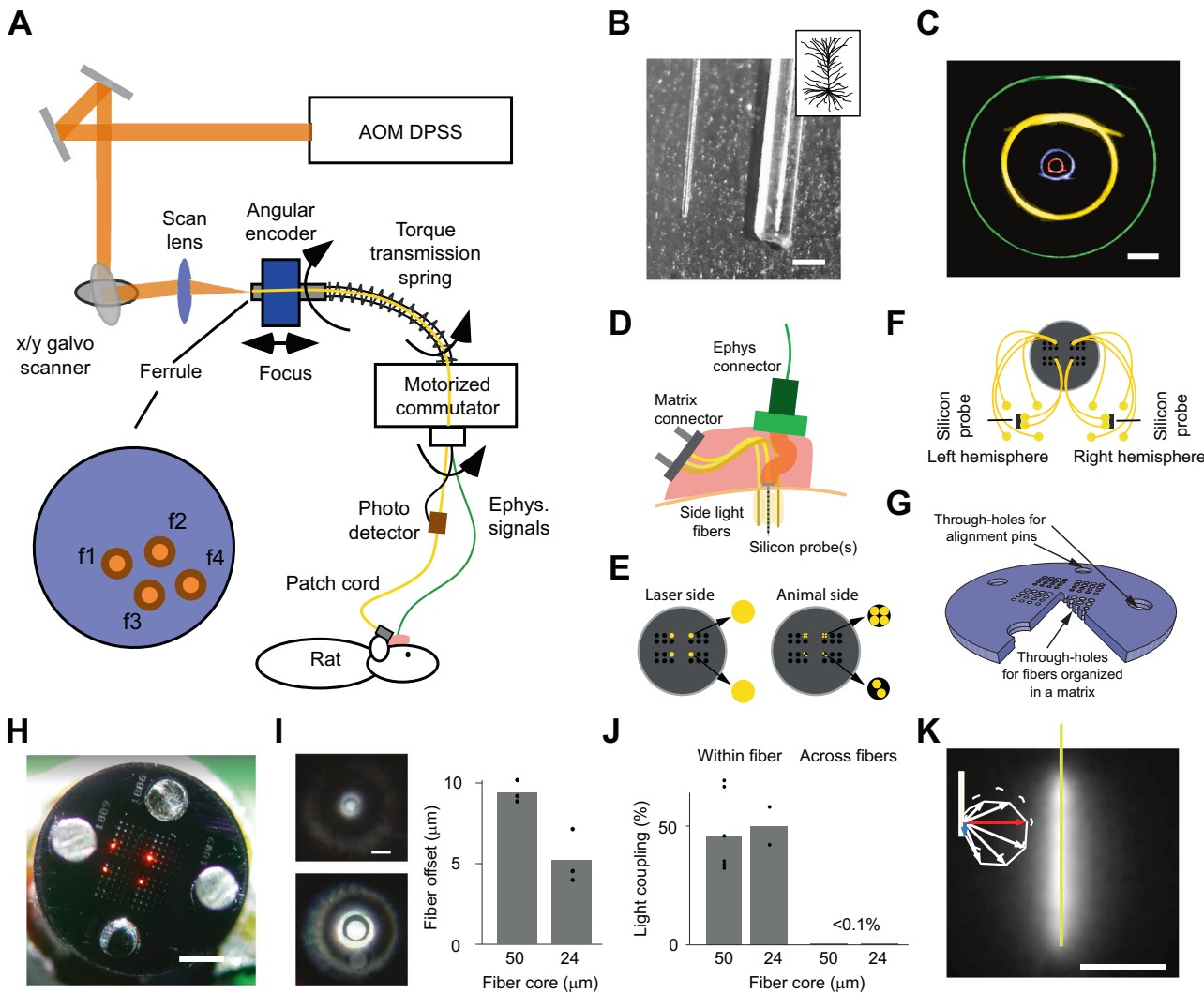

**Fig. 1 Optical framework for applications of ultrathin fibers with shaped pattern write-in in a freely moving animal. A** Optical system for automatic fiber bundle detection and optimized fiber coupling from an acousto-optically modulated (AOM) diode-pumped solid-state (DPSS) laser. Feedforward control from the ferrule angle allows the galvo scanner to track one of the selected fibers (f1, f2, f3, and f4) in a freely moving animal. **B** A 30 μm fiber next to a 230 μm fiber. Scale bar: 200 μm. The inset depicts a neuron to illustrate the size relationship. **C** Minimal bending radius for fibers with different outer/core diameter ratios: 230/200 μm (yellow), 125/100 μm (green), 65/50 μm (blue), and 30/24 μm (red). Scale bar: 2 mm. **D** Implantation schematics. The fibers (yellow), matrix connector (gray), electrode ribbon cables (orange/red), and electrode holder (green) were all cemented (pink) to the bone. **E** Coupling from 60 μm fiber (large filled yellow circle) on the laser side to two or four 30 μm fibers (small filled yellow circle) on the animal side. **F** Topological relation between implanted fibers and fiber-matrix connector. The through-hole was 75 μm in diameter, which allowed us to insert four 30 μm fibers. **G** Simplified 3D schematic illustration of the circular fiber-matrix connector plates, with a triangular cutout to illustrate the cross-section through the 380 μm-thick plate, created in silicon using microsystems processing (four 4 × 4 arrays are shown for improved visibility; the actual connector had four 6 × 6 arrays). **H** Photograph of fiber-matrix connector. Fibers were reverse-illuminated to visualize their location in the fiber-matrix connector. Scale bar: 1 mm. **I** Left: Fiber hole alignment for a 30/24 μm outer/core diameter fiber (top) and a 65/50 μm outer/core diameter fiber (bottom). Scale bar: 50 μm. Right: Quantification of the offset between two connector plates for two different hole sizes for 30 μm-diameter fiber (24 μm core) and 65 μm-diameter fiber (50 μm core). **J** Connector transmission efficiency and crosstalk between neighboring fibers in the connector. **K** Coated 2 mm fiber in milk. Scale bar: 1 mm. Note that, among the different emission directions tested (white, blue, and red arrows), the radial side-emitted component (red arrow) is larger than the axial-forward component (blue arrow). Source data are provided as a Source Data file.

significant ($p = 0.57$, $n = 5$, Student's $t$-test, slope $= -0.27 \pm 0.96$ mm$^{-1}$, SFig. 5D, E). This illustrates that, despite even emission along with the fiber, a Lambertian angular distribution is crucial for even illumination intensity in a partially scattering and absorbing tissue. The strong scattering in the brain allows the light emitted from our side-emitting fibers from the backside of the electrode to go around the electrode shank so that the light reaches the electrodes with an intensity that is 39.8% of that behind the electrode shank (SFig. 5F). Moreover, the PVR

generated by the BE probe corresponded to a light intensity that was 2.3 times that of the fiber matrix (SFig. 5G).

After 10 days of implantation, there were clearly discernible action potentials for constellations with both coated and uncoated fibers (SFig. 5H). The action potential amplitudes ranged between 100 and 200 μV, with on average of 0.77 sorted units per electrode (uncoated: $0.78 \pm 0.27$, mean ± sd, $n = 12$ sessions; coated: $0.76 \pm 0.41$, mean ± sd, $n = 7$ sessions) (SFig. 5I). Consistent with the equal recording quality, the astroglia staining decreased

**Table 1 Implanted fibers and electrodes.**

| Animal | Optical fibers | | | | Silicon probes | |
|---|---|---|---|---|---|---|
| | Right hemisphere | | Left hemisphere | | | |
| | Fiber matrix (x4) | BE probe (x2) | Fiber matrix (x4) | BE probe (x2) | R.H. | L.H. |
| 540 | NC | NC | C | C | ATLAS | BLBT |
| 541 | C | NC | C | C | ATLAS | BLBT[a] |
| 542 | C | NC | C | C | ATLAS | BLBT[a] |
| 548 | C | NC | C | C | ATLAS | BLBT |
| 549 | C | (C) | C | C | ATLAS | BLBT |
| 550 | NC | C | NC | NC | ATLAS | BLBT[a] |
| 551 | NC | C | NC | NC | ATLAS | BLBT[a] |

NC not coated, C coated, R.H. right hemisphere, L.H. left hemisphere.
[a]Large fluctuations (>1 mV) in the extracellular signal fluctuations. These fluctuations were most likely due to reuse of the ZIF-clip connector, and those probes were excluded from further analyses.
(C) = In one BE probe, the fibers were assumed to be defective because there was no PVR. The ATLAS and BLBT probes are Michigan-style silicon-based intracortical probes. The BLBT probe was produced as a part of BrainLinks-BrainTools.

**Table 2 Injection, implantation, and euthanasia dates.**

| Animal | Injection date | Implantation date | Euthanasia date | Implantation duration (days) | Expr. dur. for hist. (days) |
|---|---|---|---|---|---|
| 540 | 20 Dec 2019 | 22 Feb 2020 | 15 Mar 2020 | 22 | 64 |
| 541 | 20 Dec 2019 | 27 Feb 2020 | 15 Mar 2020 | 17 | 70 |
| 542 | 20 Dec 2019 | 27 Feb 2020 | 15 Mar 2020 | 17 | 70 |
| 548 | 23 Jan 2020 | 5 Mar 2020 | 15 Mar 2020 | 10 | 42 |
| 549 | 25 Jan 2020 | 5 Mar 2020 | 15 Mar 2020 | 10 | 40 |
| 550 | 25 Jan 2020 | 6 Mar 2020 | 15 Mar 2020 | 9 | 41 |
| 551 | 27 Jan 2020 | 6 Mar 2020 | 15 Mar 2020 | 9 | 39 |

similarly with the distance to the fiber for both the coated and uncoated fibers (SFig. 5J, K). The BE probe can thus support homogeneous illumination and high-quality extracellular recordings.

The final prerequisite for robustly aligned recordings and stimulations is the handling of light artifacts, such that extracellular action potentials can be reliably detected before, during, and after optogenetic inhibition. In three rats that had been injected with AAV2/5 carrying a hSyn-eNpHR3.0-mCherry construct, we recorded neural activity with the BE probe (Fig. 2A). Using a PVR management procedure (see methods) that can be applied after any spike-sorting scheme, we demonstrated a drastic reduction of the influence of PVR on spike count statistics (Fig. 2B).

**Ultrafast in vivo optogenetic inhibition.** Next, we analyzed the neuronal responses to laser light. We found clear optogenetic inhibition in probes that passed through tissue that contained eNpHR3.0-mCherry expressing neurons (Fig. 2C and SFig 6). The buildup of inhibition was very quick (Fig. 2D) and completed within 2.2 ms (Fig. 2E). This is faster than the time constant of halorhodospin[21]. A strong hyperpolarization will cancel spikes before the time constant has been reached (Fig. 2F) and several times faster than more physiological, channel-based or indirect inhibition[10], and it reinforces the fundamental property of optogenetics in terms of rapid silencing of genetically and anatomically defined neurons[22]. The low latency inhibition of genetically defined neurons is a prerequisite for reliable photo tagging (Fig. 2D and E). For example, an indirect effect can be ruled out for the unit shown in Fig. 2D because the inhibition was faster than the synaptic conduction between neighboring neurons, i.e., ~1 ms[23,24]. Spikes were typically eliminated for the duration of the light pulse or longer. The prolonged suppression of spikes outlasting the actual inhibition time has been described for optogenetic inhibition[25,26] and for physiological inhibition[27].

The inhibition strength approached an asymptote of ~3 mW, which corresponds to a power density of 12.5 mW/mm$^2$ (Fig. 2G). The inhibition was similar for the BE probe and fiber-matrix emission (Fig. 2H), which confirms similar PVR amplitudes for both conditions (SFig. 5G). This in turn confirms the validity of placing optical fibers behind laminar probes. Finally, it is conceivable that the pre-perturbation activity of the perturbed neurons can affect the activity during or after inhibition[28]. To this end, we divided the trials into high and low pre-inhibition spike counts for three different groups of neurons with low, moderate, and high firing rates (Fig. 2I). In general, trials with high pre-inhibition activity had higher activity after inhibition onset, and this effect was significant for neurons with moderate and high firing rates (low rate: $p = 0.98$, $n = 82$, $0.0018 \pm 0.11$ Hz, mean ± sem; moderate rate: $p = 0.0035$, $n = 82$, $0.53 \pm 0.18$ Hz; high rate: $p = 0.025$, $n = 82$, $0.46 \pm 0.2$ Hz).

**Three-dimensional stimulation and laminar recordings.** Next, we took advantage of the thin optical fibers to allow optogenetic stimulation to occur at different depths in brain tissue. To this end, we bundled 10 fibers of different lengths, resulting in an axial stimulation resolution of 500 μm and a total range of 5 mm (a 250 μm resolution and 2.5 mm range were used for the anterior implantations) (Fig. 3A). A fiber-bundle base diameter of less than 80 μm was achieved by combining fibers with diameters of 11.8 and 30 μm. The 11.8 and 30 μm-diameter fibers were polished to lengths of ~500 and 2000 μm, respectively. The 11.8 μm fiber was polished using a cooling liquid (milk or water) to avoid heat-related bending of the fibers. Approximately 70–200 fibers were polished simultaneously (Fig. 3B and SFig. 7A–D), and ~50% of those fibers had an appropriate light distribution (Fig. 3A). The fibers on the unpolished side were arranged next to each other with a fiber center-to-center distance of ~80 μm (SFig. 7E, F). We implanted six such depth probes bilaterally in a mouse brain (Fig. 3C). The connectors were grouped and

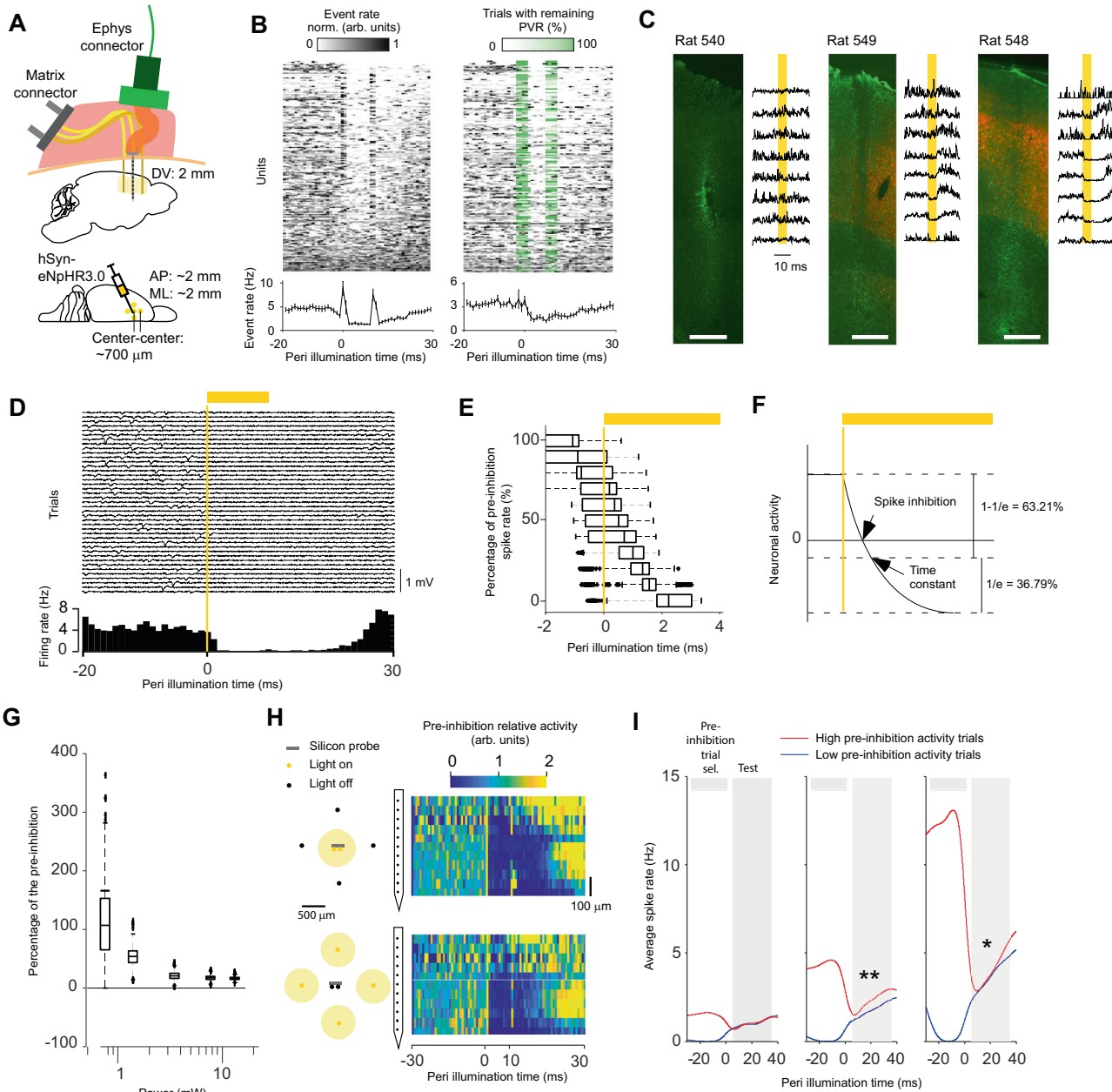

**Fig. 2 Ultrafast optogenetic inhibition in freely moving animals. A** Injection, stimulation, and recordings in M1. **B** Activity of sorted units without (left) and with (middle) PVR management. The PVR management algorithm first subtracts an estimate of the PVR and then estimates in which trials the subtraction was incomplete. This remaining PVR trial percentage (green) is then used to weight the average across trials for each time point (lower left and right panel). The data shown are from animals 540, 548, and 549. **C** Relationship between opsin expression (red) and inhibition strength. To quantify the overall extracellular response, spikes were detected at a threshold of −30 μV on a per-channel basis without PVR management (PVR removed for visibility). GFAP immunostaining (green) was used to identify the BE probe location. Scale bar: 500 μm. **D** Single-trial extracellular traces for 40 inhibitions from one electrode channel (upper panel). Histogram of spikes detected with a threshold of −30 μV (lower panel). **E** Latency at different percentages of the pre-inhibition spike rate bootstrapped across all sorted units. $n = 212$ sorted units over two animals. Boxplots: central mark indicates the median, bottom and top edges refer to the 25th and 75th percentiles of the bootstrapped data. **F** Illustration showing that the spike inhibition latency can be shorter than the time constant of the hyperpolarizing function. **G** Percentage of pre-inhibition spike rate as a function of the total light power that exits the side-emitting fiber. $n = 45, 45, 51, 63,$ and 63 sorted units with non-NaN-bins during stimulation for each light intensity, respectively. For a description of the Box-plot see panel **E**. **H** Comparison of inhibition in vivo for the BE probe (top) and the fiber matrix (bottom). Spikes were detected at a threshold of −30 μV on a per-channel basis without PVR management. **I** Inferring neuronal activity after inhibition onset from pre-inhibition activity. The units were divided into three groups corresponding to low (left, $p = 0.98$), moderate (middle, $p = 0.0035$), and high (right, $p = 0.0249$) average firing rates. *$p < 0.05$, **$p < 0.01$. Two-sided $t$-test. No adjustments for multiple comparisons. Source data are provided as a Source Data file.

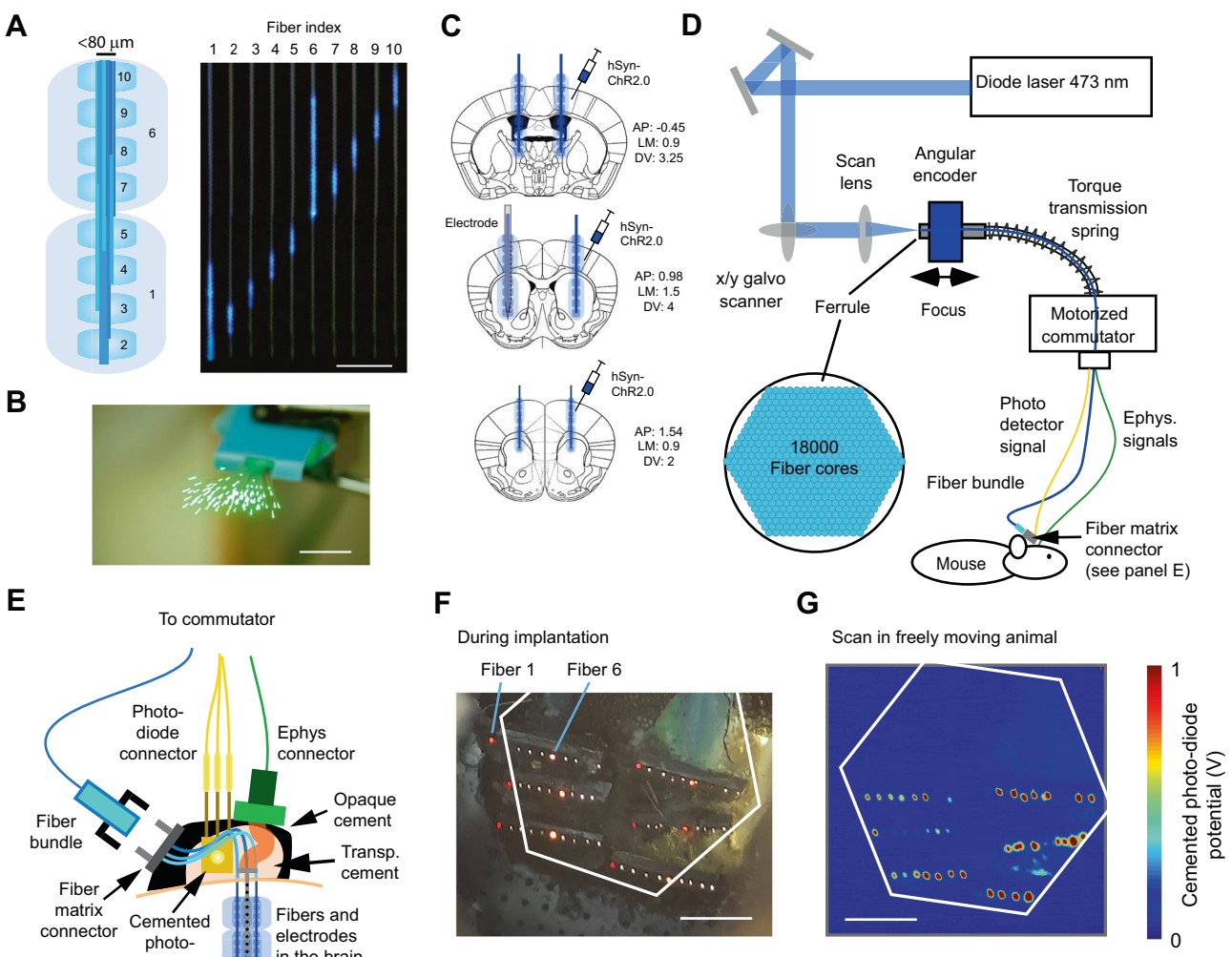

**Fig. 3 System for three-dimensional optogenetic stimulation and laminar recordings in freely moving mice. A** Fiber-bundle probe used in the experiments for light emission at different depths in the brain tissue. Scale bar: 1 mm. Schematic side view (left panel); sequentially illuminated fibers lead to defined depth illumination (right panel). **B** Multifiber polishing holder and ~70 simultaneously polished fibers illuminated with green light for visualization. Scale bar: 4 mm. **C** Bilateral ChR2 injection locations, fiber bundle implantation locations, and BE probe implantation illustrated in three exemplary coronal sections. **D** Optical set-up for flexible three-dimensional optogenetic stimulation (fiber-bundle shown as a blue line and a photodetector signal cable shown as a yellow line) and laminar recordings (ephys-cable shown as a green line) in a freely moving mouse. A fiber bundle transmits the galvo scanning position to the animal. The fiber bundle (Schott imaging fiber bundle; see methods) was 2 m long and attenuated the light by 75%. **E** Implantation schematic. The locations of the fibers in the fiber-matrix connector were detected using an implanted photodetector. The fibers emit stray light that can be detected by the photodiode in the transparent cement. This photodiode signal was sent via the commutator to the recording system. **F** Connector behind an aligning glass plate during implantation. To make the fibers visible, a white light source was directed toward the mouse skull. The deep 30 μm fibers (Fiber 1) exhibited a red emission with a low intensity (leftmost fiber in each row). The superficial 30 μm fibers (Fiber 6) exhibited a yellowish emission with a higher intensity (middle fiber in each row). The 11.8 μm fibers absorbed a higher percentage of the white light above the brain and therefore emitted white light (scale bar: 500 μm). Some fibers were not covered by the fiber bundle because the connector was aligned by manual adjustments of the original silicon plate. For the fiber bundle, the silicon plate is not necessary, and the connector on the animal side can have hexagonal constraining walls (3D-printed or CNC machined) to ensure alignment with the hexagonal fiber bundle. **G** Readout of the implanted photodiode during X/Y galvo scanning (scale bar: 500 μm). Voltage is linearly mapped to the jet pseudocolor scale in MATLAB. Source data are provided as a Source Data file.

attached to the mouse skull such that they could be scanned and individual fibers could be identified using a fiber bundle (Fig. 3D, E). This fiber bundle allowed us to assemble the connector during implantation and therefore eliminated the constraints inherent in a fixed connector (Fig. 3F). The scanned image plane was thus transmitted to the animal connector, and the individual fibers were detected using a photodiode that was submerged in dental cement (Fig. 3G). The fiber bundle allowed only minimal light leakage into neighboring fibers because the attenuation between the fibers was close (~2.5%) to that predicted (1%) by the patch cord with an integrated photodiode (SFig. 7F).

The fibers were implanted in the primary motor cortex, striatum, and thalamus of two mice. These areas had received virus injections with a channelrhodopsin construct three weeks earlier. The fiber bundle that ended in the left striatum was attached to the backside of a 32-channel silicon probe (similarly to the BE probe described above). With this BE probe, sorted units at different depths could be differentially activated (Fig. 4A). The 30 μm fiber was more effective than the 11.8 μm fiber in generating neuronal spiking (Fig. 4B). To achieve a behavioral effect with the 11.8 μm fiber, the effect of multiple fibers had to be pooled (Fig. 4C). In contrast, a single 30 μm fiber generated

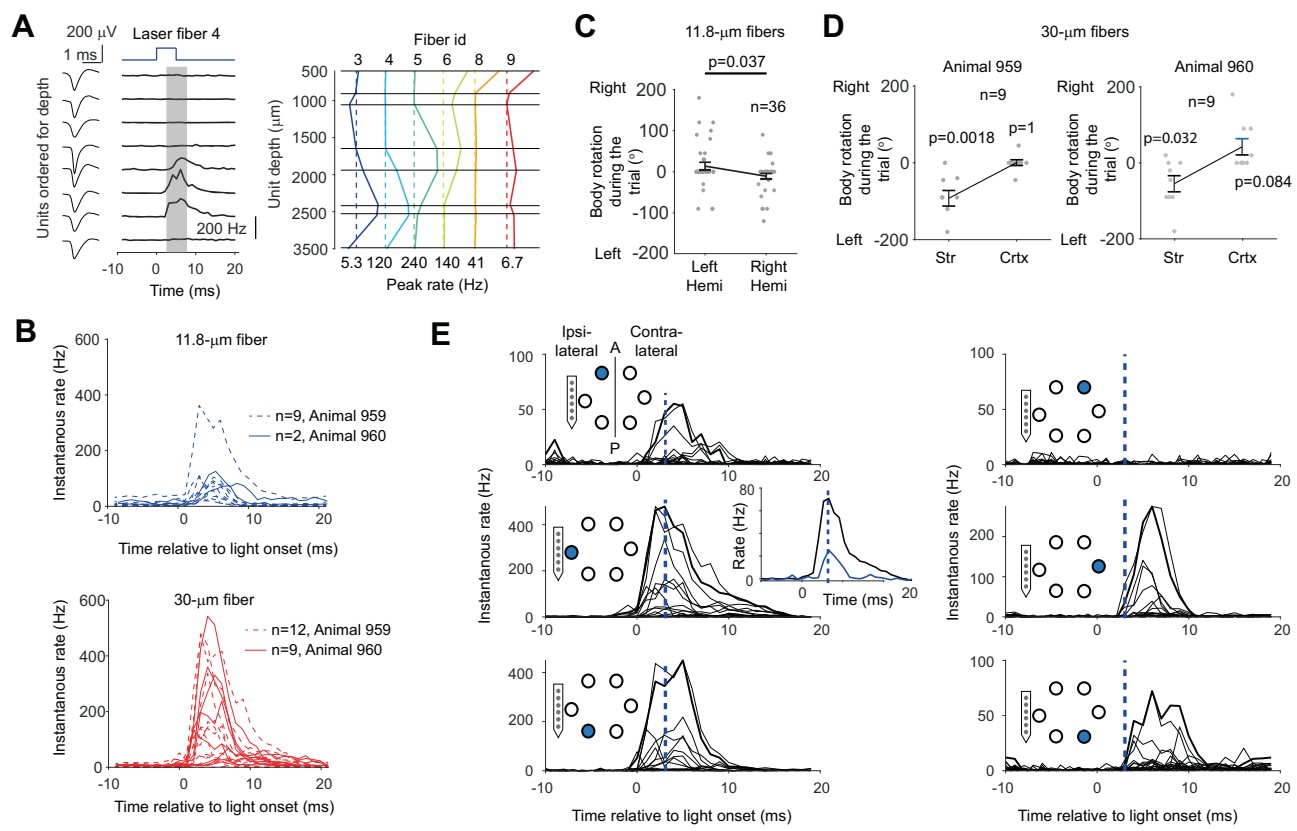

**Fig. 4 Three-dimensional optogenetic stimulation and laminar recordings in freely moving mice. A** Depth-resolved activation of sorted units (spike shapes: left column). For each unit, the firing rate was calculated 3–8 ms after light onset (the gray shaded area in the middle column), and the background firing rate preceding light onset was subtracted to quantify the light-evoked rate modulation for different units, depths, and fibers (right column, responses for a certain fiber is coded with a pseudocolor). Fiber 7 was nonfunctional and was removed for clarity. The emitted intensities from the 11.8 and 30 µm fibers were estimated before implantation to be in the ranges of 0.5–1 and 1.5–3 mW, respectively. **B** Units with significant activations for 11.8 (upper, blue lines) and 30 µm (bottom, red lines) fibers from two animals. The number of units that responded to the light is denoted by *n*. The total numbers of units were 12 for animal 960 and 25 for animal 959. **C** Optogenetic effect for left and right optical fiber on behavior measured in body rotations. In both hemispheres, all two, three, four, and five fibers were stimulated in nine trials each, resulting in 36 trials in total for each hemisphere. The amount of body turning was estimated through visual inspection across four cameras. Error bars denote the standard deviation of the mean. Two-sided *t*-test, not corrected for multiple comparisons. **D** Behavioral effect for depth-resolved stimulation in cortex (Crtx) versus striatum (Str) for the 30 µm fibers in the right hemisphere. The behavior was estimated according to panel **C**. The number of trials is denoted by *n*. Error bars denote the standard deviation of the mean. Two-sided *t*-test, not corrected for multiple comparisons. **E** Cortical recordings during the stimulation of six different superficial 30 µm fibers (stimulated fibers are indicated by a blue circle) organized bilaterally at three different positions along the anterior (A)–posterior axis (P). To rule out that longer latencies for the stimulation on the contralateral hemisphere (right column) were due to light leakage to the recorded hemisphere, we calculated the peak latency (vertical dashed blue line) for the minimum light intensity in the recorded hemisphere that generated a response (blue response curve in the middle inset). Source data are provided as a Source Data file.

reliable behavioral responses, thereby allowing a within-probe comparison of deep versus superficial stimulation (Fig. 4D). Finally, we tested whether the multi-areal fiber approach could be combined with extracellular recordings to probe neural connectivity. For fibers ipsilateral of the BE probe, the latency of the neural responses was similar to that generated by the fibers of the BE probe (Fig. 4E). In contrast, for fibers contralateral to the BE probe, there was a consistent latency shift of ~3–4 ms. This is consistent with interhemispheric connectivity latencies. Thus, this approach shows that it is possible to conduct simultaneous laminar recordings and multifiber stimulations, 3D optogenetic stimulation, connectivity inference, and behavioral quantification in a freely moving animal.

## Discussion
The stimulation and recording framework presented here allows for maximally temporally controlled interrogation of neural

circuits in freely moving animals. Although all-optical approaches, such as two-photon (2 P) imaging combined with holographic 2 P photostimulation, allow optogenetic manipulation of neuronal activity with the near-single-cell resolution, 2 P setups typically require the animal to be head-fixed. In addition, the choice of opsin is constrained by the need to maintain a sufficient spectral distance between the excitation wavelength of the indicator and the opsin. Our approach was specifically designed to work in freely moving animals and produce electrophysiological recordings as a readout. Therefore, it not only offers a higher temporal resolution than the indirect readout of activity during calcium imaging but also enables free choice of the excitation wavelength. Our BE-based approach thus fills a gap in the extant methodology by offering temporally precise readouts of neuronal activity in freely moving animals with the possibility of spatially refined photostimulation.

The spatial resolution in our approach is achieved by placing the fibers in any desired XYZ position around the electrode.

Although our fibers have a small diameter of less than 12 μm (core diameter 8 μm) and individual fibers can be separated by this distance along or orthogonally to the shank, the spatial resolution will nevertheless be drastically limited by light scattering in brain tissue. Although micro-LEDs have the same light-scattering limitation, they can be integrated between electrode sites rather than at the edge of the shank. However, this increased spatial resolution comes with the disadvantages that micro-LEDs are limited to one wavelength per LED and that they cause electromagnetic interference artifacts via their stimulation current, thus limiting the temporal resolution. Even though electromagnetic artifacts can be made smaller than the spike detection threshold[17], they will nevertheless interfere with the spike shape and therefore with spike sorting. Finally, irrespective of how such a selective optical stimulation is achieved, the stimulation specificity cannot be completely verified with extracellular recordings because of their limited spike detection horizon. Thus, unless recordings can be obtained from all neurons that are affected by the light[29], one must rely on imaging methods for which the plane of focus can be changed to verify the stimulation specificity[30]. We therefore have combined laminar recordings with fiber stimulation, because fibers allow great flexibility in multi-areal stimulation, at a scale of hundreds of micrometers to several millimeters, without causing electromagnetic artifacts.

Previous efforts to optimize fiber-based experiments were based on fiber-coupled laser diodes and tapered fibers. The advantage of a tapered fiber is the ability to sweep the center of illumination continuously along the fiber; this is not possible with our approach because we use discrete fibers to target different depths. The disadvantages of a tapered fiber are the relatively thick diameter, the stiffness, the induced pear-shaped illuminated volume, and the angular input coupling, which does not allow automatic scaling up to multifiber coupling. The need for multifiber and multi-area stimulation increases as more opsins are soma-targeted, such that one can avoid stimulating en-passant axons and dendrites[31,32].

Stark et al. pioneered the use of multiple fiber-coupled laser diodes together with silicon probes[33]. This approach was integrated into a silicon chip that included a wavelength combiner for multiple wavelengths[34], which is convenient for recently developed multi-wavelength constructs, such as BIPOLES[32].

Our approach extends the Stark approach in that we use thin fibers that do not require hazardous etching and in that the thin fibers are flexible. Furthermore, our approach extends previously used approaches[31,32] by means of side-emitting fibers, back emission for laminar electrodes, depth-resolved stimulation, an optical commutator that eliminates the need for an implanted diode laser for each fiber, and a fiber bundle that eliminates the need to have contacts with a predefined layout. The fiber bundle renders the optical interface more flexible than the electrical interface. This flexibility allows a high degree of freedom during multi-areal implantations. In fact, there is no need for tailored patch cords for each type of experiment, because the fiber bundle is by definition adaptable to the particular spatial configuration of the fiber connector on the animal. Thus, although our optical system is complex, it has never been easier to prepare and perform complex implantations.

Multiple laminar probes can record the spiking activity of 100 or more single units in practically any combination of brain areas. This multi-areal approach is indispensable for studying inter-areal communication. Given the optical properties of brain tissue, optical fibers seem to be ideal for use with laminar recordings, allowing optogenetic perturbations of this communication. Furthermore, by exploiting the interface between optical fibers and fiber bundles, we have produced a flexible optical interface that puts minimal constraints on electrode implantation. This simplifies not only the targeting of multiple areas but also the targeting of areas defined by fluorescence while avoiding superficial blood vessels. The thinness of the fibers allows simultaneous extracellular recordings in all targeted areas to confirm the effectivity of optogenetic intervention and to control for changes in excitability[35,36] The truly simultaneous recordings obtained during stimulation are of crucial importance for optically tagging neuronal subtypes and for understanding how trial-by-trial variability in neural activity changes the context of perturbations of specific neuronal subtypes in freely moving animals.

## Methods

**Producing the fiber connector using MEMS technologies**. The process, performed in the cleanroom facility of the Department of Microsystems Engineering (IMTEK) of the University of Freiburg, employed standard 4 inch, double-sided polished silicon (Si) wafers with a thickness of 380 μm. Each wafer front side (FS) was first coated with a 3.5 μm-thick silicon oxide (SiO$_x$) layer using plasma-enhanced chemical vapor deposition (PECVD). Next, a 10 μm-thick positive resist AZ9260 (Microchemicals GmbH, Ulm, Germany) was applied using UV photolithography. The resist served as a masking layer in the subsequent reactive-ion etching (RIE) process that was used to pattern the SiO$_x$ layer on each wafer FS. Each wafer FS was then etched to a depth of 200 μm using an advanced Si etching process known as the Bosch process (or deep reactive-ion etching (DRIE)) in an inductively coupled plasma (ICP) etching system.

Next, the FS masking layer was removed by wet chemical etching using 5% hydrofluoric acid (HF). The wafers were then thermally oxidized to a layer thickness of 500 nm. This silicon dioxide (SiO$_2$) layer conformally covered the wafer surface, i.e., the etched wafer front as well as the planar rear side. This layer served as an etch stop inside the recesses etched from the wafer FS. Then, a 2 μm-thick PECVD SiO$_x$ layer was deposited on the wafer rear side (RS), serving as a masking layer and patterned by UV lithography and RIE. Subsequently, DRIE was used to etch the wafers to a depth of ~150 μm. The wafers were then fixed on a handle wafer using a thermally conductive wax-like material (i.e., Cool Grease, 7016, AI Technology Inc., USA) to keep their temperature low, and were subsequently etched through using the DRIE process, stopping at the thermally grown SiO$_2$ layer. The handle wafer was removed by dissolving the Cool Grease layer. The circular connector plates were finally released from the wafer by applying torsional forces to break the suspension arms. The 500 nm-thin SiO$_2$ layer inside the through-holes cracked easily during the subsequent cleaning steps and did not block the circular openings of the connector plate.

To accommodate process tolerances, we created 1 mm through-holes (for the guide and anchoring pins) with diameters between 994 and 1015 μm that increased in increments of 3 μm (the large holes were made to fit guide pins with a maximal diameter of 1 mm and a minimal diameter of 0.995 μm). Four identical square arrays with 6 × 6 smaller circular through-holes were arranged in the middle of the connector plate. Within an array, the through-holes were positioned at a center-to-center distance of 150 μm. The through-holes for the guide and anchoring pins on the wafer front side (FS) had diameters varying from 22 to 39 μm in increments of 1 μm and from 49 to 83 μm in increments of 2 μm, respectively. Accordingly, across four wafers, the 30 μm fiber fitted into the 31 μm hole (but not into the 30 μm hole), and the 65 μm fiber fitted into the 67 μm hole (but not into the 65 μm hole), thereby confirming the predictable accuracy of the manufacturing process. The diameter of each hole on the wafer rear side (RS) was enlarged by 10 μm to compensate for potential misalignment between the FS and RS lithography process steps described below. In the case of the larger though-holes with a diameter of 1 mm, which were used for the alignment pins, we added disk-shaped dummy structures to maintain similar lateral dimensions of the structures to be etched across the entire wafer. This feature was implemented to achieve comparable etch rates of the smaller fiber through-holes and the larger trenches defining the disk shape and the large through-holes.

**General fiber handling**. Because the 11.8 and 30 μm fibers are practically invisible on a white background, the workshop desk was covered with a black poster board (TB5, Thorlabs, Germany, Bergkirchen). The 11.8 μm fiber practically floats in the air; we therefore recommend turning off the air conditioning during handling. To make the fibers easier to handle, we added a layer of Kwik Cast (WPI, Germany, Friedberg) to the grabbing surfaces at the tip of forceps. Finally, because it is difficult to pick up a fiber lying on a flat surface with bare fingers, we used a clay stick to pick up fibers. This clay stick was made by squeezing a small amount (~0.1 ml) of Paddex (UHU, Germany, Bühl/Baden) through a 1 mL syringe so that a small amount of sticky clay protruded from the hole in the tip of the syringe.

**Polishing side-emitting fibers**. The side-emitting fibers were created by gradually polishing away the cladding from a thin optical fiber (30 μm cladding, 24 μm core, NA = 0.86, S17, Lifatec, Germany). To evaluate the polishing, the other end of the

fiber was attached to a light source. The cladding was removed using a 6 μm diamond polishing paper (LF6D, Thorlabs, Germany, Bergkirchen). The polishing paper was attached to a disc with a diameter of 60 mm rotating at ten turns per second. For optimal polishing results, one should add a few drops of milk onto the rotating disc. A thin plastic sheet can then be used to evenly spread the liquid across the wheel. The milk dissipates heat, which is crucial to prevent bending when one polishes 11.8 μm fibers (8 μm core, 11.8 μm core + cladding, and 13 μm core + cladding + acid-soluble glass, NA = 0.38, Schott Fiber Optics, SAP number 1119562). The fiber was gently pushed toward the rotating polishing paper using a cotton swab, the wooden end of which was covered by a rubber coating (Kwik Cast, WPI, Germany, Friedberg). To polish different sides of the fiber, it was fixed inside a tube, which in turn could be rotated. After the polishing, the light escaped the fiber with a higher intensity at the front than at the back. A true side-emitting fiber with equal intensity at the front and back and maximal intensity at the sides was achieved by applying a thin diffusing layer of nail polish diluted by 1:4 with acetone (NailPolish Misslyn, nail polish, white, 90, Innovative Cosmetic Brands, Karlsfeld, Germany). Once the fiber was finished, it was cut to a length suitable for handling (~80 mm). In a serial fashion, additional fibers could be made by pulling out the fiber from the rotating tube, because the total fiber length was ~1 m.

Multiple fibers were polished in parallel using the rotating disc. Approximately 100–200 fibers (11.8 or 30 μm in diameter and 60 mm long) were gently bundled together and superglued together on one side. The fibers on the other side were cut to the same length with scissors. This bundle was then put inside a pencil-like holder with an LED light source (Oslon SSL 80 grün auf Square 10, OSRAM, Munich Germany) such that all of the fibers in the bundle emitted light. It is important that the LED light source be positioned at a distance (~8 mm) from the fiber bundle inlet. This distance makes the angular spread of the coupled light similar to that of a laser source. The fiber bundle was held in place with a rubber-coated paper clip. The pencil holder was then held manually so that the fibers rested on the polishing disc. The rubber-coated cotton swab tip was then used to push the fibers onto the rotating polishing disc. When the polishing was done on one side, the fiber bundle was manually turned 180°. Once the polishing was done, the individual fibers typically stuck to each other and had to be separated. The remaining milk was removed using water, and acetone was used to separate the fibers.

The variation in the length of the side-emitting segment is ~±200 μm, depending on the skill level of the polisher and the time invested in the polishing process. The reproducibility of the polished lengths was similar for single and multifiber polishing. The emission angles in SFig. 3B are reproducible as long as the nail polish is diluted 1:4 with acetone. The accuracy and reproducibility can also be increased using multifiber polishing followed by post-hoc selection of appropriate fibers. For the depth-resolved probes used in the mouse experiments, fibers were polished simultaneously, after which only the best 50% were used for the final probe. The evenness and length of the side-emission segment were determined by a laser source with a 473 nm wavelength coupled to a standard 200 μm fiber.

The emission at a certain angle was calculated, in Eq. (1), from the gamma-uncorrected intensity values summed across all of the pixels of a manually selected region of interest delineating the polished region of the fiber at a certain angle to the camera.

$$\text{Emission at angle} = \sum_{xy \in \text{ROI(angle)}} I_{xy}^{\text{angle}} \qquad (1)$$

The images, $I_{xy}$, were captured with a Samsung S5KGM2 sensor and measured in $25° \pm 5°$ increments. The intensity varies according to $I = I^0 \cos(\text{angle})$, where the angle is zero radially from the fiber and 90° axially/along with the fiber, consistent with the properties of a Lambertian-like diagram.

The total light power that exits the side-emitting fiber was estimated from the intensity given by the patch cord (with the integrated photodiode) multiplied by the efficiency of the fiber-matrix connector. The emission density can be estimated by assuming that the light must pass through a cylindrical area that surrounds the fiber. Given a maximal light power emitted by the fiber of 10 mW and a 2 mm side-emitting segment, the power density at the fiber surface (a radius of 15 μm gives a cylinder area of $0.188 \text{ mm}^2$) is estimated to be $53 \text{ mW/mm}^2$. At a distance corresponding to the maximal extracellular spike detection radius (a radius of 100 μm gives a cylinder area of $1.26 \text{ mm}^2$), the power density is estimated to be $7.96 \text{ mW/mm}^2$.

**Assembling the implant for the bilateral BE probe in the rat**. The implant was assembled on a 3D-printed substrate. This substrate served to hold fibers, electrodes, and their connectors during implantation with a single stereotactic holder. As the implantation proceeded, parts of the substrate were melted away with a standard cauterizer (Bovie, WPI, Germany, Friedberg). The 3D-printed substrate had a two-step staircase. On the longest staircase, the electrodes and fibers to be implanted first were located. Once those had been implanted, their plastic could be removed, and the electrodes and the fibers at the second staircase could be implanted.

Fibers, electrodes, and their connectors were attached to the substrate as follows. To hold the fibers and BE probes for one hemisphere (one staircase), a honeycombed arrangement of 19 guide tubes (i.e., two guide tube layers away from the center guide tube), with an inner diameter of 200 μm, an outer diameter of 350 μm (TSP200350, Polymicro, Phoenix, AZ, USA), and a length of 3 mm were

glued to each staircase. The BE probes consisted of two side-emitting fibers attached to the backside of the laminar electrodes (two 30 μm-diameter fibers fit behind a 75 μm-wide electrode). The fibers were attached to the electrodes using superglue (UHU, Bühl/Baden, Germany) at the base of the laminar electrodes. Then, the BE probes were inserted into the middle guide tube on the respective staircase, and four fibers were inserted into four of the most peripheral guide tubes at a distance of 600 μm from the BE probe in a star-like fashion. To protect the fibers during implantation and to form a multifiber ribbon cable, they were coated with rubber up to a distance of 10 mm from the guide tubes.

Before the fibers were inserted into the silicon plate, we added two 1 mm-diameter, 3 mm-long low-precision steel pins in the guide pin holes that were not used for the high-precision pins. This ensured a rigid connection between the silicon plate and the rest of the connector. The four fibers on the respective staircases were inserted into one 65 μm hole. The two fibers coming from one BE probe were inserted into another 65 μm hole, and two additional dummy fibers were added. The dummy fibers prevented unnecessary heat buildup at the connector junction. The fiber length between the silicon plate and guide tubes was set to 35 mm. The same procedure was repeated for the other staircase/guide tube bundle. The fibers were attached to the silicon plate using a heat-curable epoxy. To this end, small drops of epoxy (using a 33 gauge wire) was added to all of the fibers from both sides of the plate to prevent undercutting during polishing. A small heat coil with a diameter of 1 mm and a length of 5 mm (200 μm-diameter Nichrome wire with a current control to achieve an orange glowing color), attached to a pen-like grip, was used to heat the air in the vicinity of the epoxy. A heat gun could not be used because that would have blown away the small parts. Before the plate was polished, the remaining bare 25 mm fibers were coated with rubber to finalize the multifiber ribbon cable. A standard surgical stereo microscope with ×40 magnification was used to inspect the polishing.

Next, the guide pins were added to the silicon plate. An indent was polished at one end of the guide pins to aid their fixation in the contact. The other end was inserted into a fixture that ensured the right spacing and a 90° angle of the guide pins. The fixture with the two guide pins was coated with Vaseline to prevent epoxy from clogging the fixture. This assembly was coupled to the polished silicon plate with the fibers. A 3D-printed connector sleeve was slid over the rubber-coated fibers and down to make contact with the guide pins. Finally, the connector sleeve, fibers, silicon plate, and guide pins were epoxied together using slow-curing epoxy (End-fest, UHU, Germany, Bühl/Baden).

**Assembling the patch cord with an integrated photodiode for a bilateral BE probe in a rat**. The patch cord with an integrated photodiode relies on a small amount of light that escapes the fiber and passes through the protective tube to be detected by a linear photodetector on the outside of the patch cord. To this end, we used a yellow tube with an inner diameter of 600 μm and an outer diameter of 900 μm (FT900Y, Thorlabs, Germany, Bergkirchen).

The assembly of the patch cord began with insertion of the four fibers (AS50/60IRPI, coating of 65 mm polyimide, LEONI Fiber Optics, Schierschnitz, Germany) into the protective tube. Then, the fibers were gently inserted into the four holes of the silicon plate that corresponded to the holes used by the implant. For stability, two 1 mm-diameter, 3 mm-long low-precision steel pins were inserted in the guide pin holes that were not used for the high-precision pins. Before securing the pins with heat-curing epoxy, we ensured that all fibers had the same length before entering the protective tube. Identical fiber lengths prevent individual fibers from forming knots or loops as the protective tube is slid toward the silicon plate. Next, the silicon plate was polished and coupled to two high-precision guide pins that were inserted in a Vaseline-covered fixture. Two 3.5 mm tubes were cut from a 17 gauge injection needle. Those were filled with Vaseline and slid over the high-precision guide pins that extended from the patch cord side. Then a 3D-printed patch cord sleeve was glued to the tubes, silicon plate, anchor pins, and tubes. The tubes added robustness to the connector. We used the same patch cord for all sessions involving freely moving rats, and the power transmission for the patch cord was measured after all of the recordings were done (End-fest, UHU, Germany, Bühl/Baden).

The fibers on the other side of the patch cord were coated with heat-curing epoxy and inserted into four guide tubes with an inner diameter of 70 μm and an outer diameter of 200 μm (TSP075200, Polymicro, Molex, USA). This bundle was inserted in a metal ferrule with an outer diameter of 2 mm that was filled with heat-curing epoxy. Note that the arrangement of individual fibers is flexible because each fiber is addressed by the galvo scanner. Finally, the metal tube was gently heated to facilitate an even distribution of epoxy in the metal tube. Care was taken to slowly heat up the epoxy (so that the curing would emerge after ~20 s) to prevent the formation of heat-induced cavities between the fibers, which in turn would hamper proper polishing.

**Assembling the laser system**. The laser system (see Fig. 1A) was designed to achieve high-intensity multichannel pulsed optogenetic inhibition. A high-intensity laser source is favorable as it can compensate for the relatively large coupling losses that occur with small optical fibers. Here we used one 100 mW DPSS Bragg-modulated laser (594 nm, Mambo, Cobolt, Solna, Sweden) for the inhibition experiments and a diode laser (473 nm, BrixX, Omicron) running at 100 mW for the excitation experiments. The side-emitting fibers spread out the photon current

over a larger area, thus preventing optical burning of the tissue. This in turn opens up for more efficient optogenetic inhibition. Furthermore, the 10 ms light pulses used here cause minimal heat buildup.

The optical axis passed through a galvo scanner with two mirrors (8315KSM40B, Cambridge Technology, Bedford, MA, USA). The galvo mirror had a high-power servo amplifier unit and deflected the beam into one of the fibers within 100 μs. From the galvo mirror, the light passed through a scan lens (LSM03_VIS, Thorlabs, USA), which in turn focused the light into one of the fibers in the patch cord. The ferrule of the patch cord was located in an angular encoder with 32,768 steps (7700D05000R00S0175N, Gurley Precision Instruments, Troy, NY, USA) that was supported by two ball bearings. A stepper motor stage (Mini25, Luigs Neumann, Ratingen, Germany) that could be moved along the optical axis served to focus the light at the ferrule in the optical commutator. The patch cord was turned by a motorized electrical commutator (MEC) (ACO32, TDT, Alachua, FL, USA) when the animal turned, and the angular encoder ensured that the galvo scanner always positioned the beam on the appropriate fiber. A metal spring transferred the torque from the motorized commutator to the angular encoder. In this way, the optical axis of the laser system could be at a 90° angle to the rotating axis of the MEC. The patch cord passed through the MEC and down to the animal. The MEC commutated the extracellular signals from 48 electrodes and two light intensity signals from the patch cord with the integrated photodiode. All signals were integrated via a QPIDe IO card (Quanser, Markham, Ontario, Canada) and MATLAB scripting (Mathworks, Natick, MA, USA).

**Animals**. All animal procedures were approved by the Regierungspräsidium Freiburg, Germany. In this study, we used seven male Sprague Dawley rats (400 g, Janvier) that were implanted at the age of eight weeks. Three to four animals were pair-housed in type-4 cages (1500U, IVC typ4, Tecniplast, Hohenpeißenberg, Germany) with a humidity between 45 and 65% and a temperature between 20 and 24 degrees, before implantation, and the animals were single-housed after the implantation in type-3 cages (1291H, IVC typ3, Tecniplast, Hohenpeißenberg, Germany) with a humidity between 45 and 65 % and a temperature between 20 and 24 degrees, under a 12 h reverse light-dark cycle (dark period from 8 a.m. to 8 p.m.; the period during which training and experiments were conducted).

In this study, we used two male C57BL/6 mice. The animals were housed (cage type: Blueline Type 1284 L, Tecniplast, Hohenpeißenberg, Germany) with a humidity between 45 and 65% and a temperature between 20 and 24 degrees, under a 12 h light-dark cycle (light period from 8 a.m. to 8 p.m.; the period during which training and experiments were conducted).

**Animal surgery and virus injection: rats**. The animals were initially anesthetized by isoflurane inhalation, followed by intraperitoneal injection of 75 mg/kg ketamine (Medistar, Holzwickede, Germany) and 50 mg/kg medetomidine (Orion Pharma, Espoo, Finland). The animals were put into a transportation container covered with an opaque cloth to facilitate the anesthetization. Once the animals were anesthetized, they were positioned in a stereotaxic frame (David Kopf Instruments, Tujunga, CA, USA), and their body temperature was maintained at 36–37 °C using a rectal thermometer and a heated blanket (FHC, Bowdoin, USA). The animals were kept anesthetized using ~2% isoflurane and 0.5 l/min $O_2$. For pre-surgery analgesia, we subcutaneously (s.c.) administered 0.05 mg/kg buprenorphine (Selectavet Dr. Otto Fischer GmbH, Weyarn/Holzolling, Germany). Every other hour, the animals received an s.c. injection of 5 mL isotonic saline. The moisturizing ointment was applied to the eyes to prevent them from drying out (Bepanthen, Bayer HealthCare, Leverkusen, Germany). The skin was disinfected with Braunol (B. Braun Melsungen AG, Melsungen, Germany) and Kodan (Schülke, Norderstedt, Germany). To perform the craniotomy, the skin on the head was opened along a 2 cm-long incision using a scalpel. The exposed bone was cleaned using a 3% peroxide solution. For virus injections, craniotomies were drilled bilaterally extending from −1 to +3 mm in the anterior–posterior direction and from +0.5 to +1 mm in the lateral–medial direction relative to Bregma. We injected a 1 μl viral vector (AAV-hSyn-eNpHR3.0-mCherry-WPRE, UNC Vector Core, North Carolina, Chapel Hill) at a rate of 100 nL/min into the respective subareas using a 10 μl gas-tight Hamilton syringe (World Precision Instruments, Sarasota, FL, USA). To minimize the reflux of the injected volume, we left the injection needle in the tissue for 10 additional minutes before slowly extracting it from the brain. To protect the brain during virus expression, the craniotomy was covered with bone wax. The surgical site was closed around the implant by interrupted sutures using 5-0 silk (SMI AG, St. Vith, Belgium).

**Animal surgery and virus injection: mice**. The mice were first briefly anesthetized with a mixture of oxygen and isoflurane (up to 5%) and then injected with buprenorphine (0.05 mg/kg, s.c.). Subsequently, the animals were fixed in a stereotactic mount under a maximum of 3% isoflurane. The anesthesia was supplemented with oxygen via a rodent mask and maintained by isoflurane (1.5–3%). The ability to add isoflurane via the rodent mask at any time permitted quick intervention in case an animal's depth of anesthesia was reduced. The depth of anesthesia was determined by the inter-toe reflex and respiratory rate and was regulated by varying the concentration of the inhalation anesthetic. The eyes were protected with eye ointment (Bepanthen eye and nose ointment), and a heating pad ensured

that the animals were kept warm. During the surgical procedure, the animals were injected with subcutaneous Ringer's solution every 2 h to prevent fluid loss.

For virus injections, craniotomies were drilled bilaterally (coordinates: position 1: LM: 0.95, AP: −0.46; position 2: LM: 1.5, AP = 0.98; position 3: LM: 0.9, AP = 1.54). For position 1, we injected 200 nL of viral vector (AAV-hSyn-ChR2-EYFP, UNC Vector Core, North Carolina, Chapel Hill) at a rate of 100 nL/min into the respective subareas using a 10 μl gas-tight Hamilton syringe (World Precision Instruments, Sarasota, FL, USA). For position 2, we injected 100 nL at five different depths: 0.5, 0.9, 2, 2.7, and 3.4 mm. For position 3, we injected 100 nL to depths of 0.4 and 0.9 mm. To minimize reflux of the injected volume, we left the injection needle in the tissue for 10 additional minutes before slowly extracting it from the brain. To protect the brain during virus expression, the craniotomy was covered with bone wax. The surgical site was closed around the implant by interrupted sutures using 5-0 silk (SMI AG, St. Vith, Belgium).

**Animal surgery and BE probe implantation: rats**. The initial steps of BE probe implantation were identical to those of the virus injections (see above). Self-tapping skull screws (J.I. Morris Company, Southbridge, MA, USA) for reference for extracellular recordings were placed over the cerebellum. For BE probe implantation, craniotomies were drilled bilaterally from −2 to +5 mm in the anterior–posterior direction and from +1 to +3 mm in the lateral–medial direction relative to Bregma. The BE probes with the surrounding fiber matrix were implanted bilaterally, beginning with the right hemisphere. To this end, the degradable staircase implant holder (see "Assembling the implant for the bilateral BE probe in the rat") was held with a stereotaxic manipulator arm and positioned so that the lowest stair-step was located over the right hemisphere. The electrode and multifiber ribbon cable faced anteriorly, and the 3D-printed substrate faced posteriorly. The right BE probe shank was inserted 2 mm into the cortex, with care taken to ensure that no blood vessel was punctured by the BE probe or the fiber matrix. Kwik-Cast (WPI, Sarasota, FL, USA) was applied to the craniotomy to protect the brain from the dental cement (Paladur, Kulzer GmbH, Hanau, Germany) that was applied to anchor the first stair-step to the bone. After 5 min, when the dental cement had cured, a cauterizer was used to cut the degradable implant holder just above the first stair-step to dissociate the implant holder from the implanted stair-step. Next, the remaining left stair-step was positioned over the left hemisphere by moving the stereotaxic manipulator arm. The BE probe was inserted to a depth of 2 mm, Kwik-Cast was applied, and the stair-step was anchored to the bone. After 5 min, the dental cement had cured, and the left stair-step could be dissociated from the degradable holder. At this stage, only the fiber-matrix connector and ZIF-clip connectors were held by the stereotaxic manipulator arm. Those connectors were dissociated from the holder using the cauterizer and attached to the skull with dental cement. Finally, the fiber and electrode ribbon cables were protected with dental cement.

**Animal surgery and BE probe implantation: mice**. The initial steps of BE probe implantation were identical to those of the virus injections (see above). At each injection site, a small cut in the dura mater was made to allow the BE probe and fibers to be inserted into the brain tissue. The ten-fiber probe (with or without silicon probe) was held at its fiber connector with an orthogonally attached 25 gauge needle. The needle tube was held by a flattened crocodile clamp. This allowed the fiber probe to be turned around all axes, except around its own axis, and thereby facilitated axial insertion into the brain tissue. Once the fiber was inserted, it was attached to the bone with a tiny drop of superglue, followed by a natron-saturated water solution for instant curing. The needle tube was then released from the flattened crocodile clamp so that the fiber connector side could be attached to the fiber connector holder. Finally, the needle tube was released from the fiber connector side by drilling away the plastic that joined the fiber bundle connector and the needle (see "making a staggered fiber bundle" below). The fiber connector holder was cemented to two skull screws and the bone above the cerebellum. Because we attached multiple fiber bundles (six pieces) to the fiber connector holder, we had to ensure that the polished side of all corresponding connectors terminated in the same plane. Therefore, we used the magnets in the fiber connector holder to hold a 200 μm-thick glass slide that served as a physical stop for the individual fiber bundle connectors. Finally, a drop of gel superglue was used to secure the connector to the holder.

When all six connectors were attached, their backsides were covered with black-pigmented dental cement to block the light. During the fiber scanning, the light must reach the photodiode via the fibers and not through the spaces between the fibers. The photodiode is cemented to the bone where the fibers pass through the craniotomies; this cement does not contain any black pigment. Finally, a thin layer of black-pigmented dental cement covered the complete implant; that is, behind the fiber connector, there is a "pocket" of more transparent dental cement through which the light can escape the fibers and reach the photodiode.

**Making a staggered fiber bundle**. To manufacture the staggered fiber bundle, we made a miniature fixture. Central to this fixture are ten small-diameter tubes (100/50 μm outer/inner diameter) that are slightly angled relative to each other to create a fan-like structure. This structure guided the individual fibers toward each other while ensuring a certain spacing between them. To facilitate insertion of the fibers

into those narrow holes, each tube was polished (at the entry of the side with the larger spacing between the tubes) at an angle to make the entry hole oval-shaped (and therefore longer). The guide tubes were ~5 mm long.

The procedure began with picking up one sidelight fiber using the clay stick and inserting the fiber into the leftmost guide tube, with the polished side going in first. As the fiber came out the other side, it was pulled out over a printed scale (500 μm between the marks) to the correct length. The fiber was then fixed to a small pillow of Paddex (UHU, Germany, Bühl/Baden) behind the entry side of the guide tubes. This procedure was then repeated for the remaining nine fibers. The procedure was the same for the 11.8 and 30 μm fibers.

Once all of the fibers were the appropriate length, they were attached to a silicon plate (B × W × L = 100 × 800 × 2000 μm) with superglue. The silicon plate was located below the fibers and roughly 1 mm in front of the exit holes of the guide tubes. Although the guide tubes ensured that a certain distance was maintained between the fibers, the distances between the fibers on the silicon plate were manually adjusted before they were glued. This was facilitated by placing a tiny drop of body lotion on the plate; the body lotion did not dry out and did not attract the fibers to one another. When the fibers were arranged equidistant from each other (based on visual inspection), a drop of superglue was placed on both sides of the body lotion. It is important not to use too much superglue for this purpose because it makes the connector thicker, which in turn allows fewer connectors to fit within the field of view of the fiber bundle.

Once the ten fibers had been secured to the silicon plate, they were bundled using Kwik Cast (WPI, Germany, Friedberg). The Kwik Cast was applied, starting at the silicon plate and continuing to approximately 2 mm from the shortest fiber. The 2 mm gap was necessary to glue the fiber bundle to the bone during implantation. Finally, all of the sidelight fibers were bundled using nail polish diluted by 1:4 with acetone (NailPolish Misslyn, nail polish, white, 90, Innovative Cosmetic Brands, Karlsfeld, Germany).

The fiber bundle connector was then polished using 6 μm-grain diamond polishing paper (LF6D, Thorlabs, Germany, Bergkirchen) and 1 μm-grain aluminum oxide polishing paper (LF1P, Thorlabs, Germany, Bergkirchen). To polish the relatively small fiber bundle connector, it was temporarily glued to a 25 gauge needle via a 0.3 × 1.5 × 6 mm piece of plastic. The plastic piece was glued orthogonally to the backside of the silicon plate (the fiberless side) so that the 6 mm side protruded orthogonally to the silicon plate and the 1.5 mm side was orthogonal to the 2 mm side of the silicon plate. The 25 gauge needle was then glued to the rectangular plastic piece. This configuration was used for polishing and also to hold the fiber bundle during implantation. Once the fiber bundle connector was glued to the fiber holder on the animal, the needle could be released from the bundle connector by drilling away the plastic.

**Data acquisition and spike sorting using KiloSort**. Extracellular signals were bandpass-filtered, amplified, and digitized using INTAN (Intan Technologies, Los Angeles, California) two-head stages (RHD2132) that were integrated into zero-insertion-force clips (ZD32, Tucker Davis Technologies) for the bilateral BE probe rat experiments and one-head stage (RHD2132) that was integrated into zero-insertion-force clips (ZD32, Tucker Davis Technologies) for the unilateral BE probe mouse experiments. To maximize the animals' comfort, we suspended the ultrathin INTAN cable by an ultralight spring with a 1.5 mm diameter.

The extracellular recordings were filtered with a 7.5 kHz low-pass cutoff (third-order Butterworth) and a 0.1 Hz high-pass cutoff (first-order Butterworth) and sampled at 30 kHz, after which the signal was digitally high-pass filtered with a 1.0 Hz cutoff (first-order Butterworth). The PVR was removed off-line (see "PVR management procedure" below) using Signal processing toolbox and Statistics toolbox in matlab. Spike sorting was done using KiloSort with default parameters. Units with a clear PVR response 1 ms after the light onset were not used for further analysis.

**Calculating the intensity ratio between the BE probe and the fiber matrix**. To calculate the light intensity ratio at the electrodes between the BE probe fibers and the fiber matrix, we equalized the PVR for the fiber matrix and BE probe fibers by changing the relative intensity (given by the patch cord with an integrated photodiode) for those two-fiber configurations. For example, by fixing the intensity for the fiber matrix, we could modify the intensity for the BE probe fibers such that the PVR became equal for the two configurations. Using the two resulting intensities, we could calculate the ratio for the BE probe and the fiber matrix. To find matching intensities, we first estimated the range in PVR amplitudes for which they were linearly related to the light intensity. The relation between light intensity and PVR was linear up to ~1000 μV, and for high light intensities, the PVR saturated at +6000 μV. For each electrode, we estimated the time point after the laser onset for which the maximal intensity in the fiber matrix caused a PVR that was closest to 1000 μV. If the largest PVR for a given channel was less than 1000 μV, the time point of the corresponding maximal PVR amplitude was used. This time point was used to determine the intensity of the BE probe that generated the most similar PVR amplitude for the given channel. The intensity ratio was calculated as the maximal intensity for the fiber matrix divided by the latter intensity of the BE probe.

**PVR management procedure**. Because the light source was aligned with the electrodes, the Becquerel-induced light artifacts were relatively large. The strategy adopted was to eliminate as many of the PVRs as possible, and if a PVR could not be removed, it was marked as "black out period"/"not a number" (NaN) in order to facilitate further processing. In short, PVRs were first decreased by subtracting PVR templates based on the applied light intensity. Second, a heuristic was used to estimate time points of the remaining PVRs; these time points were "blacked out" for spike detection (the intensity of the green intervals in Fig. 2B denotes the percentage of trials with remaining PVRs). Finally, spike templates that were sorted during non-stimulated intervals were fitted to threshold crossings during stimulation and were accepted as true spikes if their errors were less than the standard deviation and if they did not contain a blackout interval. The algorithm thus identified PVR waveforms and did not classify them as neuronal units.

To remove PVRs, data snippets were cutout beginning at 30 ms before the light pulse onset and ending 30 ms after the light pulse offset. Light onset and offset were defined according to the half-maximum latency of the light intensity in the patch cord with the integrated photodiode. The resulting data snippets of 70 ms were classified into eight groups according to the four different fiber channels and the two different electrode shanks. PVR removal was performed individually for each of the eight groups. First, the data were high-pass filtered; they were then supersampled by a factor of four to increase the temporal resolution. The transient PVRs associated with light onset and offset were then minimized. We tested several methods, such as linear and nonlinear regression, principal component analysis, nonlinear principal component analysis, and template matching. The best result was achieved with template matching. Because of the large number of pulses (on the order of 500 pulses) it was possible to tailor each template for each pulse. A template was created by averaging the 20 pulses with the smallest mean squared error to the pulse of interest. Averaging 20 pulses resulted in a decrease in the signal amplitude (spike amplitude) of approximately 5%. Because the duration of each pulse varied, the templates for the onset and offset transients were calculated individually. To avoid discontinuities in the data, the templates for the onset and offset were calculated with an overlap of 3.3 ms (100 samples at 30,000 samples per second) between the onset and offset. Linear interpolation was used to join the onset and offset templates across this 3.3 ms overlap (for 100 samples).

We employed a multistep procedure to detect remaining PVRs. To capture the remaining high-frequency PVRs, we derived the signal in time. The remaining PVRs were concentrated where the PVR had its strongest temporal derivative, which was ~260 μs after laser onset and lasted for another 300 μs. Because errors from PVR removal can be both positive and negative, we rectified the derived signal. To improve the PVR estimate, we averaged the rectified signal across trials as follows. Because the success of the PVR removal depended on the laser intensity, the trials were sorted according to the light intensity given by the patch cord with the integrated photodiode, and a moving-window filter was run across the trials for each time point. The moving-window filter averaged the rectified values across seven trials, with the exception of the trial that had the largest rectified value. The largest rectified value was omitted to minimize the influence of spikes. If the resulting signal for a certain trial and for a certain time point was more than four times greater than the standard deviation of the pre-stimulation period, it was regarded as a PVR, and the corresponding data point in the PVR-minimized data was replaced by NaN. These data were referred to as the result of the PVR management procedure.

Because it is unclear how KiloSort[16] (or any other spike-sorting routine) handles optical PVRs, we chose to run the spike sorting on the PVR-free periods between the pulses by concatenating those periods to one matrix. The number of clusters was assumed to be twice the number of channels. After the spike sorting, we ran the automatic curation. Sorted units with a negative spike amplitude larger than 400 μV and three spike amplitudes larger than 20 μV were omitted because of their unphysiological properties. The remaining sorted units were matched to spike events in the PVR-managed data. This was done as follows. For a given trial, all of the channels were normalized to have standard deviations equal to one. The channels were sorted according to the linear layout of the probe. Then, the smallest negative event was chosen (and marked with a NaN so that it would not be chosen again) and compared to the waveforms of the sorted units. To this end, three channels centered on the negative event were subtracted from the corresponding channels of one of the sorted units' waveforms. The unit yielding the smallest squared error was selected. If this error was less than one (the standard deviation of the data), the waveform was classified as an action potential. This procedure was repeated for the next spike and stopped when no event remained that was more than four times larger than the standard deviation.

To minimize PVR-related spike bias, an event detected in the spike-matching procedure was regarded as invalid if the corresponding spike window (−20 to 40 samples surrounding the event) contained a NaN. A separate data structure was used to keep track of NaNs. After the spike-matching process was completed, the regions containing NaNs were expanded 40 samples back in time and 20 samples forward in time. This served to label all periods in which no action potential could occur (because the spike window was not allowed to contain a NaN).

The photovoltaic response strongly depended on the type of electrode. The ATLAS probe responses are an order of magnitude larger than those of the IMTEK probe (see SFig. 5B). As a result, for the IMTEK probe, spikes could be detected reliably (Fig. 2D). It should also be noted that we used sharp light onset according to a step function with a 0.1 ms slope. A ramp over the course of 1 ms drastically

decreases the PVR. However, even in this case, artifact removal is necessary for the detection of spikes (see https://www.ucl.ac.uk/neuropixels/training/2021-neuropixels-course, 3.7 - Combining Neuropixels with optogenetics - Maxime Beau (UCL)). Fortunately, we were able to detect action potentials with high fidelity in both mice and rats without implementing light ramps.

**Histology**. Brains transferred from a sucrose solution were attached to the cooling block of a microtome with Tissue Tek (Sakura Finetek, Fisher Scientific, Germany). The slices were transferred to vials containing phosphate-buffered saline (PBS) with 0.01% sodium azide. For antibody staining, selected slices were washed for $3 \times 10$ min in PBS on a rotary shaker at room temperature. The slices were blocked and permeabilized for 30 min (PBS 0,01 M/Triton 0,4%/BSA 5%, Sigma Aldrich, St. Louis, MO, USA) on the rotary shaker. The first antibody (dilution 1:1000, anti-GFAP, PA5-16291, Thermofisher, Waltham, MA, USA) was applied overnight at 4 °C (PBS 0,01 M/Triton 0,2%). The slices were washed for $3 \times 10$ min in PBS on the rotary shaker at room temperature. The second antibody (dilution 1:250, Alexa 488 goat anti-rabbit, A11034, Thermofisher, Waltham, MA, USA) was applied for 2–3 h (PBS 0,01 M/Triton 0,2%). Finally, the slices were washed for $3 \times 10$ min in PBS on the rotary shaker at room temperature, mounted, and stained with DAPI (Vectashield, H-1200-10, Burlingham, CA, USA). The slices were imaged with a Zeiss microscope using a ×5 objective. The exposure time for DAPI, Alexa 488, and mCherry were 40, 800, and 70 ms, respectively. Shading correction was done in Fiji (based on ImageJ from NIH, Bethesda, Maryland, USA) using the BaSiC plugin[37]. Stitching was done in Fiji using the Grid/Collection plugin[38].

**Statistical procedures**. Double-sided *t*-tests were conducted to assess statistical significance in small-sample comparisons, assuming normal distributions. Statistics for the inhibition latency were calculated by bootstrapping the peri-inhibition time histograms of the spikes of sorted units with 1000 repetitions.

**Reporting summary**. Further information on research design is available in the Nature Research Reporting Summary linked to this article.

## Data availability
A minimal dataset in terms of the depth-resolved stimulation generated in this study have been deposited on GitHub: https://github.com/Optophys/Multichannel_optogenetics. The data that support the findings of this study are available from the corresponding authors upon request. Source data are provided with this paper.

## Code availability
The code for analyzing the data that support the findings of this study is available from the corresponding authors upon request.

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

## Acknowledgements
We would like to thank Ofer Yizhar for comments on an earlier version of this manuscript and Julian Ammer for advice on the comparison of our approach to two-photon imaging. We would like to thank Michael Dargie at Schott Fiber optics for kindly providing us two-fiber bundle samples. This work was supported by the Bernstein Award 2012 (01GQ2301), the BrainLinks-BrainTools Cluster of Excellence (EXC 1086), the Deutsche Forschungsgemeinschaft (DFG) via Grants DI 1908/3-1, DI 1908/11-1 and DI 1908/6-1, and the ERC Starting Grant OptoMotorPath (338041), all to I.D., and the Research Innovation Fund to D.E.

## Author contributions

D.E., P.R., and I.D. conceived the study and wrote the manuscript. D.E. designed the optical hardware with help from A.T. and A.S. D.E. performed the in vivo experiments. D.E. and P.R. designed the silicone connector plate. D.E. designed the multifiber-depth and fiber-bundle approach. P.R. manufactured the silicone connector plate. D.E. performed the data analysis. A.S., M.A., and D.E. perfused the animals. D.E. performed the histology, with help from M.A. K.S. designed, fabricated, assembled, and characterized the BLBT probes. D.E. performed the light measurements in the mice, with help from B.C.

## Funding

## Competing interests

The authors declare no competing interests.
