## [Peer Review File · Nature Communications]

Multichannel optogenetics combined with laminar recordings for ultra-controlled neuronal interrogationREVIEWER COMMENTS

Reviewer #1 (Remarks to the Author):

The work by Eriksson et al reports a method to implement optical light delivery by thin optical fibers around a shank for extracellular recording in free-moving animals.

The work is well written, pleasant to read and I like the methodology described by the authors.

The integrated multi-functional connector/commutator, together with the possibility to implant multiple fibers with a shank for multisite extracellular electrophysiology, represents a novel approach to realize arrays of multiple implants. As the authors mention in the highlights, the approach is enabled by a set of developments, building up a system that does not have the axial resolution provided by μ LED probes or tapered fibers, but can interface with cortical columns providing for optical delivery and extracellular electrophysiology for the entire cortical depth in free-moving animals.

One of the keys of this work is the back-end part for assembling together the different elements, an aspect that is often neglected in the literature and that makes the work by Eriksson et al very interesting for the community working on implantable systems for optogenetics.

I'm therefore very supportive on the innovation introduced by the system, although there are some points that the authors should address to improve the manuscript with reference to improve quality of presentation and tailor some of the claims to be more specific:

11
SEP Major

- To this reviewer opinion, one main point is that it is not completely clear how the electrophysiology shank is aligned with the fibers, e.g. how the implant looks like on the animal's head. Can the authors add a picture on this respect or a three dimensional representation of it, and give a more detailed description?
- From supplementary figure 3 it is clear that the radial emission direction is generated mainly by the diffusive coating and that the coating changes the length of the emission segment (SFig3 E-G). Therefore the final emission geometry is generated by both the diffusive properties of the coating as well as by its refractive index, which slightly modifies the guiding properties of the waveguide. This should be mentioned in the manuscript.
- How are the arrows computed in SFig 3B and Fig 1i? Unless I missed it in the long methods section, I didn't find a description for this, and it would be crucial if the authors would like to claim that the emission approaches a Lambertian-like diagram.
- Can the authors provide an estimation of the power density (in mW/mm^2) emitted by the fibers? Coupling efficiency are reported in percentage, but the light spreads in long fiber segments and for the described experiment this would be a very important parameter for the reader to know. I understand the estimation can be experimentally challenging on this very long emitting segment, but the authors could just measure a portion of the emitting fiber and try to give an estimation of it.
- Some of the claims in the last paragraph are too general and must be reformulated being more specific, for instance:
 - (i) the presented framework allows for "maximally controlled interrogation of neural circuits". It is for sure a great system, but it allows controlled interrogation of neural circuits only until a certain extent, that should be better defined by the authors.
 - 11
SEP (ii) "we have enabled this by rendering fiber-based experiments compatible to, and as flexible as, extracellular recordings". Extracellular recordings have depth resolution and single unit targeting, which is absent for the optical part.
 - 11
SEP (iii) "This allows any desired combination of fibers and electrodes" is also too general, the limitations of the spatial constraints for fibers and electrodes placements are relevant also in the case of the system proposed by the authors.

- A paragraph on the limitations of the technique must be included, comparing it with the capability of other approaches also from the point of view of downsides.

Minor

- The first time I read the highlights/introduction/abstract, I was caught by the "cell-sized Lambertian side-emitting optical fiber" wording, and I thought the authors referred to single-cell optical stimulation. Later on in the text it was then clear that they referred to the size of the fibers.
- Is the term "active patch cord" a standard nomenclature? From the working "active" I would have expected the patch cord to contain emitting material
- SFig 7F, the inset should be bigger
- Probably better avoid referring to yellow light, and just mention the wavelength.
- Line 116: "leading to a maximum of 100 individually addressable fibers per square millimeter in the ferrule (SFig. 1C)", I'm not sure how Supplementary Fig1C shows this, is the figure reference correct?

Reviewer #2 (Remarks to the Author):

This paper reports a compact assembly technique of multiple side-emitting optical fibers with silicon probes for simultaneous electrophysiology and optogenetic inhibition in behaving animals. Large-scale recording and optogenetic intervention from cell-sized multichannel light sources are crucial to decipher and understand functional connections between neurons and brain functions, as the authors stated. This paper assembles two thinned optical fibers on the backside of silicon probes (BE probes, or back emission probes) and four additional side-emitting fibers (fiber matrix) located in four corners surrounding the BE probe (Fig. 2H, and SFig. 4A). This is an interesting approach; however, the novelty of implementation should be further clarified and the compelling in-vivo experiments that will support the claim of technical merits should be presented.

Major:

1. The novelty of the proposed approach needs to be clarified.
 - a. The attachment of thinned optical fibers onto a silicon probe was not new (Stark, JNP 2012) and the side-emitting fibers can be customized by introducing specific scatterers or fluorescent additives into the fiber core, or creating asymmetries in cladding geometry or refractive index. In this work, one side of cladding layers was removed by a polishing plate. Is the novelty of this work on manufacturing, attaching and assembling the side-emitting thin (30um) fibers?
 - b. Or is the novelty claimed on the optical system for coupling four fibers from an AOM DPSS laser or a silicon-micromachined fiber matrix connector as an enabling technology? Can the LEDs or LDs be simply coupled to the end of four optical fibers (Stark, JNP 2012; Kampasi, Microsyst Nanoeng 2018), which will not need any complex optical system?
2. From Fig.1A and D, it seems that there are only four fibers can be selected for illumination: one pair for the BE probe and the fiber matrix in left hemisphere and the other pair for right hemisphere. So in one hemisphere, there are only two possible selections of illumination control: close to the silicon probe and away in four corners (Fig. 2H). Is there any good utility of this configuration in specific neuroscience studies?
3. In Fig. 2H, optogenetic inhibition effects on neural responses were similar for two different illuminations: one by the BE probe and the other by the fiber matrix. If there are no distinguishable responses, then what is the utility of implanting multiple fibers? A simple flood exposure from a single fiber would do the same effect.
4. To fully demonstrate the utility of the proposed multichannel optogenetic systems, unique and compelling neuroscience experiments should be presented that could not be achieved by conventional or other state-of-the-art optoelectrodes.

Minor:

1. In Fig. 1A, what is f5 for? Also, indicate where the fiber matrix connector (Fig. 1D,E) is located in Fig. 1A.
2. In Fig. 1D, more detailed explanation is needed how four fibers can be coupled from one though

hole.

3. In Fig. 1E, more explanation is needed to clarify why multiple through-holes are needed and how the fibers are coupled as shown in Fig. 1D.

4. In SFig. 2B and C, put a scale bar. BS should be RS in SFig. 2B.

5. In SFig. 3B, the detailed explanation of how to characterize the emission angle is needed.

6. Manufacturing of side-emitting fibers: How accurately can the polished line length be controlled in terms of micrometers? How reproducible is the manufacturing of emission angles and polished lengths?

7. In SFig. 5B, it is not clear what the family curves are. Need more explanation.

8. Photo voltaic response of the silicon probe is so severe that it can be used to profile the spatial resolution of optical illumination from the side-emitting fibers. But it would be non-ideal for opto-electrophysiology. How does this affect temporal resolution and accuracy of spike sorting and local field potential?

9. In SFig. 5F, the BE (back emission) probe gives illumination of light not only behind the electrodes but also in front of electrodes. How do you distinguish which neurons are optically stimulated? What is the spatial resolution of your optogenetic stimulation? In most optoelectrodes, illumination is confined in front of the probe and sometimes illumination intensity is controlled to stimulate a small volume at the distal end of illumination sites.

10. In Fig. 2H, it illustrates both BE probe and fiber matrix have concentric illumination profile. Is it true? Have you characterized the illumination profile around the side-emitting fiber in 360 degrees? The purpose of the side emission is to provide uniform illumination around the fiber in all directions? If the side-emitting fiber has directionality, then how do you align it when you assemble the fibers to give the maximum illumination toward the silicon probe?

Reviewer #3 (Remarks to the Author):

In this manuscript, Eriksson et al. demonstrated a fiber probe that combined Lambertian side-emitting optical fibers and silicon probes for simultaneous optogenetic stimulation and neural recording. The demonstrated ability of this probe is potentially useful for neuroscience studies, and this reviewer found the ability to individually address the location of light emission with a x/y galvo scanner particularly innovative. It is worth noting that this capability is compatible with behavioral neuroscience experiments in freely moving animals, facilitated by a motorized commutator and an angular encoder that tracks one of the selected fibers despite the rotation of the ferrule. Despite these advances, this reviewer found it difficult to track down important parameters and detailed information related to the experimental implementation of this technology. Additionally, the authors are encouraged to revise the language to improve readability. The following comments are suggested for the authors to address during the revision:

1. It is unclear how the electrophysiological recordings were collected. The authors claimed that the developed fiber probes can achieve "high quality recordings and ultrafast multichannel optogenetic inhibition in freely moving animals." (Line 82-83) To support the statement, the authors demonstrate Figure 2 and Supplementary Figure 5. However, the authors failed to provide any details about the data acquisition and analysis methods except for a single sentence on Page 4, Line 150-152, "We recorded electrophysiological signals and applied yellow laser light in freely moving rats in sessions of 60 to 120 minutes over the course of 10 days." For example, are the original recorded traces filtered, and if so, what cutoff frequency and type of filters did the authors apply to the data? What thresholding conditions were applied for sorting the single-unit spikes and how are the spikes clustered? How many neurons can be recorded by the silicon probe per channel, and are all neurons subject to optogenetic inhibition? Fig. 2B shows the response of "units" under illumination, while it remains unclear whether these units are from the same brain region and the same channel of the silicon probe or not. In that sense, the average "event rate" shown at the end of Fig. 2B is poorly defined since neurons of different types usually have vastly different baseline firing rates, ranging from <0.1 Hz to >100 Hz. Therefore, it makes more sense to summarize the results based on each identified single unit instead of showing an average firing rate (e.g., different neurons may also show distinct behavior to illumination, depending on the efficiency of viral transduction and the selectivity of transducing different neurons types based on the promoter type).

2. It is unclear whether the bundle of fiber probes can avoid blood vessels damages. The authors

claimed that the bundle of fiber probes “allow any combination of areas to be addressed, as well as individualized designs for each animal to avoid blood vessels and to target areas” The authors may include additional immunohistochemistry assays to prove the claim. Another missing information is the distance between the Si probe and the optical fibers, and a scale bar is needed for the schematics in Fig. 2A, those in the left column of Fig. 2H, Supplementary Fig. 4A, and Supplementary Fig. 5A. This information is important since single-unit recordings in the bottom panel of Fig. 2H show significant decreased firing even when the illuminated areas are far away from the silicon probe. The authors are encouraged to refer to “Spatiotemporal constraints on optogenetic inactivation in cortical circuits” *elife* 2019 for the radius of inactivation and compare it to their measurements in Fig. 2H.

3. This reviewer cannot find the exact stereotaxic coordinates in the “Animal surgery and BE-probe implantation” method section except for a (very wide) range of locations. It is known that the neuron density and types vary significantly from different brain regions and the authors need to use a consistent brain region for rigorous scientific demonstration of their approaches.

4. In comparison to previously reported tapered fibers, the authors state that “Although the elegant tapered fibers produce a more even illumination along the electrode shank, the light is typically strongest towards the fiber tip.” However, the author did not include a quantitative comparison of the intensity profiles between their optical fibers and the tapered fibers. Please perform quantitative analysis of the intensity profiles for images shown in Fig. 1I and Supplementary Fig. 3 to substantiate this claim. Since the authors now have demonstrated individually addressed fiber output based on raster scan, can they adjust the input angle and location of illumination during the programmed scan to achieve more uniform intensity distribution without using a diffusing layer?

5. It is important to demonstrate whether the fiber probe is small and light enough to be used in freely moving animals. The authors may include additional behavioral assays to convince potential readers.

6. Typos

a. Page 3, Line 91-92: “as well as well as” should be changed to “as well as”

b. Page 4, Line 168-169: The authors wrote “SFig. 5K and K.”

c. It is recommended to change “Photo Voltaic” to “Photovoltaic”

We thank all reviewers for their valuable comments. We marked all changes with track change. Due to the number of substantial changes, we decided to highlight conceptual changes with yellow while smaller changes (e.g. grammar) are marked only in track-change.

Reviewer #1 (Remarks to the Author):

The work by Eriksson et al reports a method to implement optical light delivery by thin optical fibers around a shank for extracellular recording in free-moving animals.

The work is well written, pleasant to read and I like the methodology described by the authors.

The integrated multi-functional connector/commutator, together with the possibility to implant multiple fibers with a shank for multisite extracellular electrophysiology, represents a novel approach to realize arrays of multiple implants. As the authors mention in the highlights, the approach is enabled by a set of developments, building up a system that does not have the axial resolution provided by μ LED probes or tapered fibers, but can interface with cortical columns providing for optical delivery and extracellular electrophysiology for the entire cortical depth in free-moving animals.

One of the keys of this work is the back-end part for assembling together the different elements, an aspect that is often neglected in the literature and that makes the work by Eriksson et al very interesting for the community working on implantable systems for optogenetics.

We thank the reviewer for the positive evaluation and the supportive words. After all the hard work which has been put into this manuscript this is really rewarding.

I'm therefore very supportive on the innovation introduced by the system, although there are some points that the authors should address to improve the manuscript with reference to improve quality of presentation and tailor some of the claims to be more specific:

Major

- To this reviewer opinion, one main point is that it is not completely clear how the electrophysiology shank is aligned with the fibers, e.g. how the implant looks like on the animal's head. Can the authors add a picture on this respect or a three dimensional representation of it, and give a more detailed description?

We thank the reviewer for this comment. We have now added two schematics which depicts:

The implant with the bilateral BE-probes with surrounding fibers in the rat in panel Fig 1D (page 8, lines 310-311):

D

"D: Implantation schematics. The fibers (yellow), matrix connector (gray), electrode ribbon cables (orange/red), and electrode holder (green) were all cemented (pink) to the bone."

Regarding the alignment between electrode and fibers we added the following information in supplementary figure 4A and B (page 16, lines 464-478).

"Supplementary Figure 4. Implantation of multiple fibers and electrodes. A: The fibers (yellow) were glued (blue) directly to the backside of the silicone probe (gray) at its base. This prevents interference with the ribbon cable (orange), which is typically attached to the electrode channels on the front side of the shank. B: Photo taken during surgery showing the fiber matrix (four surrounding fibers) and BE probe (white fiber next to a grayish silicon probe) (top). The fiber electrode combination allowed superficial blood vessels to be avoided, and the thinness of the fibers led to minimal dimpling upon insertion. The BE probe, consisting of a silicon electrode next to two side-emitting fibers, is surrounded by four additional side-emitting fibers (bottom). The alignment between the fibers and the probe is achieved with guide tubes (350/200- μ m outer/inner diameters) in a hexagonal structure. C: Temporary implant holder for individual targeting of multiple brain areas with multiple BE probes and additional fibers. Scale bar: 5 mm."

For the exact coordinates we added Fig 2A:

For the new experiments with the single BE-probe and the 60 fibers in the mouse we have the new panel 3E (page 11, lines 370-373):

“Panel 3E: Implantation schematic. The locations of the fibers in the fiber matrix connector were detected using an implanted photodetector. The fibers emit stray light that can be detected by the photo diode in the transparent cement. This photo diode signal was sent via the commutator to the recording system.”

- From supplementary figure 3 it is clear that the radial emission direction is generated mainly by the diffusive coating and that the coating changes the length of the emission segment (SFig3 E-G). Therefore the final emission geometry is generated by both the diffusive properties of the coating as well as by its refractive index, which slightly modifies the guiding properties of the waveguide. This should be mentioned in the manuscript.

This is an important point and we have added it to the manuscript in the Result section (page 4, lines 143-146):

“In addition to the diffusive property, this layer has a refractive index of ~ 1.4 ,²⁰ which is closer to the refractive index of the fiber core (~ 1.5) than the refractive index of water (~ 1.33). Therefore, the diffusive coating facilitates the escape of light from the fiber.”

- How are the arrows computed in SFig 3B and Fig 1i? Unless I missed it in the long methods section, I didn't find a description for this, and it would be crucial if the authors would like to claim that the emission approaches a Lambertian-like diagram.

We have now added this information in the methods section “Polishing side-emitting fibers” (pages 23-24, lines 622-628):

“The emission at a certain angle was calculated from the gamma-uncorrected intensity values summed across all of the pixels of a manually selected region of interest delineating the polished region of the fiber at a certain angle to the camera.

$$\text{Emission at angle} = \sum_{xy \in ROI(\text{angle})} I_{xy}^{\text{angle}}$$

The images, I_{xy} , were captured with a Samsung S5KGM2 sensor and measured in $25^\circ \pm 5^\circ$ increments. The intensity varies according to $I = I^0 \cos(\text{angle})$, where the angle is zero radially from the fiber and 90° axially/along the fiber, consistent with the properties of a Lambertian-like diagram.”

- Can the authors provide an estimation of the power density (in mW/mm²) emitted by the fibers? Coupling efficiency are reported in percentage, but the light spreads in long fiber segments and for the described experiment this would be a very important parameter for the reader to

know. I understand the estimation can be experimentally challenging on this very long emitting segment, but the authors could just measure a portion of the emitting fiber and try to give an estimation of it.

This is a very good question since one of the advantages with an even light emission along an extended portion of the fiber is that the light is distributed across space which in turn reduces the heat buildup. We have now added an estimation of the power density directly at the fiber surface and at a distance corresponding to the maximal extracellular spike detection radius in the legend of Figure 2G (page 10, lines 359-364):

“The emission density can be estimated by assuming that the light must pass through a cylindrical area that surrounds the fiber. Given a maximal light power emitted by the fiber of 10 mW and a 2-mm side-emitting segment, the power density at the fiber surface (a radius of 15 μm gives a cylinder area of 0.188 mm^2) is estimated to be 53 mW/mm^2 . At a distance corresponding to the maximal extracellular spike detection radius (a radius of 100 μm gives a cylinder area of 1.26 mm^2), the power density is estimated to be 7.96 mW/mm^2 .”

- Some of the claims in the last paragraph are too general and must be reformulated being more specific, for instance:

(i) the presented framework allows for “maximally controlled interrogation of neural circuits”. It is for sure a great system, but it allows controlled interrogation of neural circuits only until a certain extent, that should be better defined by the authors.

We thank the reviewer to spot this unclear phrasing. We now added some more specific description of what the system can achieve and line out its limitations (pages 6-7, lines 247-274):

“The stimulation and recording framework presented here allows for maximally temporally controlled interrogation of neural circuits in freely moving animals. Although all optical approaches, such as two-photon (2P) imaging combined with holographic 2P photostimulation, allow optogenetic manipulation of neuronal activity with near-single-cell resolution, 2P setups typically require the animal to be head-fixed. In addition, the choice of opsin is constrained by the need to maintain a sufficient spectral distance between the excitation wavelength of the indicator and the opsin. Our approach was specifically designed to work in freely moving animals and produce electrophysiological recordings as a readout. Therefore, it not only offers a higher temporal resolution than the indirect readout of activity during calcium imaging but also enables free choice of the excitation wavelength. Our BE-based approach thus fills a gap in the extant methodology by offering temporally precise readouts of neuronal activity in freely moving animals with the possibility of spatially refined photostimulation.

The spatial resolution in our approach is achieved by placing the fibers in any desired XYZ position around the electrode. Although our fibers have a small diameter of less than 12 μm (core diameter 8 μm) and individual fibers can be separated by this distance along or orthogonally to the shank, the spatial resolution will nevertheless be drastically limited by light scattering in brain tissue. Although micro-LEDs have the same light-scattering limitation, they can be integrated between electrode sites rather than at the edge of the shank. However, this increased spatial resolution comes with the disadvantages that micro-LEDs are limited to one wavelength per LED and that they cause electromagnetic interference artifacts via their stimulation current, thus limiting the temporal resolution. Even though electromagnetic artifacts can be made smaller than the spike detection threshold,¹⁷ they will nevertheless interfere with the spike shape and therefore with spike sorting. Finally, irrespective of how such a selective optical stimulation is achieved, the stimulation specificity cannot be completely verified with extracellular recordings because of their limited spike detection

horizon. Thus, unless recordings can be obtained from all neurons that are affected by the light,²⁹ one must rely on imaging methods for which the plane of focus can be changed to verify the stimulation specificity.³⁰ We therefore have combined laminar recordings with fiber stimulation, because fibers allow great flexibility in multi-areal stimulation, at a scale of hundreds of micrometers to several millimeters, without causing electromagnetic artifacts.”

(ii) “we have enabled this by rendering fiber-based experiments compatible to, and as flexible as, extracellular recordings”. Extracellular recordings have depth resolution and single unit targeting, which is absent for the optical part.

That’s absolutely true. We added this into the argumentation above.

(iii) “This allows any desired combination of fibers and electrodes” is also too general, the limitations of the spatial constrains for fibers and electrodes placements are relevant also in the case of the system proposed by the authors.

That is also true. We now toned down this aspect by writing (page 7, lines 297-309):

“Multiple laminar probes can record the spiking activity of 100 or more single units in practically any combination of brain areas. This multi-areal approach is indispensable for studying inter-areal communication. Given the optical properties of brain tissue, optical fibers seem to be ideal for use with laminar recordings, allowing optogenetic perturbations of this communication. Furthermore, by exploiting the interface between optical fibers and fiber bundles, we have produced a flexible optical interface that puts minimal constraints on the electrode implantation. This simplifies not only the targeting of multiple areas but also the targeting of areas defined by fluorescence while avoiding superficial blood vessels. The thinness of the fibers allows simultaneous extracellular recordings in all targeted areas to confirm the effectivity of optogenetic intervention and to control for changes in excitability.^{35,36} The truly simultaneous recordings obtained during stimulation are of crucial importance for optically tagging neuronal subtypes and for understanding how trial-by-trial variability in neural activity changes the context of perturbations of specific neuronal subtypes in freely moving animals.”

- A paragraph on the limitations of the technique must be included, comparing it with the capability of other approaches also from the point of view of downsides.

We did so now by contrasting our approach with 2P-imaging and stimulation as one anchor point. Please see our response to your comment (i). Further, we added a discussion about the comparison of our approach, lamda probes, and the integrated LED-approach into silicone probes (pages 6-7, lines 258-282):

“The spatial resolution in our approach is achieved by placing the fibers in any desired XYZ position around the electrode. Although our fibers have a small diameter of less than 12 μm (core diameter 8 μm) and individual fibers can be separated by this distance along or orthogonally to the shank, the spatial resolution will nevertheless be drastically limited by light scattering in brain tissue. Although micro-LEDs have the same light-scattering limitation, they can be integrated between electrode sites rather than at the edge of the shank. However, this increased spatial resolution comes with the disadvantages that micro-LEDs are limited to one wavelength per LED and that they cause electromagnetic interference artifacts via their stimulation current, thus limiting the temporal resolution. Even though electromagnetic artifacts can be made smaller than the spike detection threshold,¹⁷ they will nevertheless interfere with the spike shape and therefore with spike sorting. Finally, irrespective of how such a selective optical stimulation is achieved, the stimulation specificity

cannot be completely verified with extracellular recordings because of their limited spike detection horizon. Thus, unless recordings can be obtained from all neurons that are affected by the light,²⁹ one must rely on imaging methods for which the plane of focus can be changed to verify the stimulation specificity.³⁰ We therefore have combined laminar recordings with fiber stimulation, because fibers allow great flexibility in multi-areal stimulation, at a scale of hundreds of micrometers to several millimeters, without causing electromagnetic artifacts.

Previous efforts to optimize fiber-based experiments were based on fiber-coupled laser diodes and tapered fibers. The advantage of a tapered fiber is the ability to sweep the center of illumination continuously along the fiber; this is not possible with our approach because we use discrete fibers to target different depths. The disadvantages of a tapered fiber are the relatively thick diameter, the stiffness, the induced pear-shaped illuminated volume, and the angular input coupling, which does not allow automatic scaling up to multifiber coupling. The need for multifiber and multi-area stimulation increases as more opsins are soma-targeted, such that one can avoid stimulating en-passant axons and dendrites.^{31,32}

Minor

- The first time I read the highlights/introduction/abstract, I was caught by the “cell-sized Lambertian side-emitting optical fiber” wording, and I thought the authors referred to single-cell optical stimulation. Later on in the text it was then clear that they referred to the size of the fibers.

We changed the terminology by shifting the optical fibers directly after the word cell-sized: ‘Cell-sized optical fiber with Lambertian side emission’. By this it should become clear that we are referring to the size of the fibers.

- Is the term “active patch cord” a standard nomenclature? From the working “active” I would have expected the patch cord to contain emitting material

The reviewer is right. The word active is misleading instead we used now the wording “patch cord with integrated photodiode”.

- SFig 7F, the inset should be bigger

We assume the reviewer means supplementary figure 5F. We have now modified the inset.

- Probably better avoid referring to yellow light, and just mention the wavelength.

We now refer to yellow and blue light by the wavelength 594nm and 473 nm, respectively.

- Line 116: “leading to a maximum of 100 individually addressable fibers per square millimeter in the ferrule (SFig. 1C)”, I’m not sure how Supplementary Fig1C shows this, is the figure reference correct?

The reviewer is right, the reference to the figure was put in the wrong context. We have now corrected for that accordingly (page 3, lines 119-121):

“Optical fibers separated with a distance of 100 μm can be addressed individually with a crosstalk of 0.3% (SFig. 1C), leading to a maximum of 100 individually addressable fibers per square millimeter in the ferrule.”

Reviewer #2 (Remarks to the Author):

This paper reports a compact assembly technique of multiple side-emitting optical fibers with silicon probes for simultaneous electrophysiology and optogenetic inhibition in behaving animals. Large-scale recording and optogenetic intervention from cell-sized multichannel light sources are crucial to decipher and understand functional connections between neurons and brain functions, as the authors stated. This paper assembles two thinned optical fibers on the backside of silicon probes (BE probes, or back emission probes) and four additional side-emitting fibers (fiber matrix) located in four corners surrounding the BE probe (Fig. 2H, and SFig. 4A). This is an interesting approach; however, the novelty of implementation should be further clarified and the compelling in-vivo experiments that will support the claim of technical merits should be presented.

Major:

1. The novelty of the proposed approach needs to be clarified.

a. The attachment of thinned optical fibers onto a silicon probe was not new (Stark, JNP 2012) and the side-emitting fibers can be customized by introducing specific scatterers or fluorescent additives into the fiber core, or creating asymmetries in cladding geometry or refractive index. In this work, one side of cladding layers was removed by a polishing plate. Is the novelty of this work on manufacturing, attaching and assembling the side-emitting thin (30 μm) fibers?

We thank the reviewer for pointing us to the relevant reference of Stark et al., 2012 which we now contrast our work with and cite. The novelty of our approach relative to that of Stark is seven fold. First, we don't have to reduce the diameters of our fibers since the fibers have a suitable diameter. Second, we don't use hazardous chemicals for manufacturing the fibers. Third, since Stark et al only etched the tip of the fiber, the remaining of the fiber has a large diameter and is therefore less flexible leading to more complex implantation scenarios. Fourth, since Stark et al, only etch the tip of the fiber it is difficult to bundle multiple fibers for stimulating different depths in the brain tissue. Fifth, the etching from one diameter to a smaller diameter result in a loss of light which in turn causes lower light power densities. Sixth, our approach introduces the possibility of making a defined length of the fiber side emitting. Seventh, we show that the fiber can be placed behind the laminar probe. Thereby we can use the whole laminar probe for extracellular recordings and still evenly illuminating all those neurons along the laminar probe. Finally in the newly added experiments in the revision round we show that it is possible to have a depth resolved illumination of different segments of the laminar probe.

We have now added this to the discussion (page 7, lines 283-296):

“Stark et al. pioneered the use of multiple fiber-coupled laser diodes together with silicon probes.³³ This approach was integrated into a silicon chip that included a wavelength combiner for multiple wavelengths,³⁴ which is convenient for recently developed multi-wavelength constructs, such as BIPOLES.³²”

Our approach extends the Stark approach in that we use thin fibers that do not require hazardous etching and in that the thin fibers are flexible. Furthermore, our approach extends previously used

approaches^{31,32} by means of side-emitting fibers, back emission for laminar electrodes, depth-resolved stimulation, an optical commutator that eliminates the need for an implanted diode laser for each fiber, and a fiber bundle that eliminates the need to have contacts with a predefined layout. The fiber bundle renders the optical interface more flexible than the electrical interface. This flexibility allows a high degree of freedom during multi-areal implantations. In fact, there is no need for tailored patch cords for each type of experiment, because the fiber bundle is by definition adaptable to the particular spatial configuration of the fiber connector on the animal. Thus, although our optical system is complex, it has never been easier to prepare and perform new complex implantations.”

b. Or is the novelty claimed on the optical system for coupling four fibers from an AOM DPSS laser or a silicon-micromachined fiber matrix connector as an enabling technology? Can the LEDs or LDs be simply coupled to the end of four optical fibers (Stark, JNP 2012; Kampasi, Microsyst Nanoeng 2018), which will not need any complex optical system?

First, a fiber and a fiber matrix connector is all that has to be implanted for stimulating the brain with our system. In comparison to Stark et al, there is no alignment required between fiber and LED or LD, there is no need to solder cables, no need to have electrical contacts, no need to encapsulate the LED around each fiber, no need to have a separate LED driver for each fiber (with our approach a single light source can be switched between multiple fibers). In fact with the fiber bundle we show that one don't even need a fiber matrix connector: an optical interface has the advantage that it does not require contacts (electrical) with a fixed spacing. Thus, the optical interface supersedes the electrical in terms of flexibility. This in turn allows maximum freedom during multi-areal implantations. In fact, one don't even need to make a special patch cord for each experiment since the fiber bundle is per definition is adaptable to the particular spatial configuration of the fiber connector on the animal. Thus, although our optical system is complex, it has never been easier to prepare and perform new complex implantations.

We have now added this to the discussion (page 7, lines 283-296):

“Stark et al pioneered the use of multiple laser-diode coupled fibers together with silicon probes⁹. This approach was integrated into a silicon chip which included a wavelength combiner for multiple wavelengths¹⁰ which comes in handy for the recently developed multi-wavelength constructs such as BIPOLES⁸.

Our approach extends the Stark approach in that we use thin fibers that don't need hazardous etching and that the thin fibers are flexible. Furthermore, our approach extends both the Stark approach and the integrated silicon-chip approach by means of side emitting fibers, back emission for laminar electrodes, depth resolved stimulation, optical commutator eliminating the need for an implanted diode laser for each fiber, and fiber bundle eliminating the need to have contacts with predefined layout. The last point means that the optical interface supersedes the electrical in terms of flexibility. This in turn allows a high degree of freedom during multi-areal implantations. In fact, there is no need to make a dedicated patch cord for each type of experiment since the fiber bundle is per definition adaptable to the particular spatial configuration of the fiber connector on the animal. Thus, although our optical system is complex, it has never been easier to prepare and perform new complex implantations.”

2. From Fig.1A and D, it seems that there are only four fibers can be selected for illumination: one pair for the BE probe and the fiber matrix in left hemisphere and the other pair for right hemisphere. So in one hemisphere, there are only two possible selections of illumination control: close to the silicon probe and away in four corners (Fig. 2H). Is there any good utility of this configuration in specific neuroscience studies?

This is a good question and we regret that it's not clear from the manuscript why we chose this fiber configuration. We decided to have this configuration in order to test the feasibility of having an optical fiber behind a laminar probe as it allows us to compare the BE-fiber emissions with the surround fibers which are not blocked by the laminar probe. Please further note that we added experiments with 60 fibers in this revision process (see Fig 3) and our response to comment 4.

We have now motivated the fiber matrix more clearly (page 4, lines 151-156):

*"To test the validity of this back emission (BE) probe, it was surrounded by four additional fibers (hereinafter referred to as the "fiber matrix") at a distance of 700 μm (S**Fig. 4B**). Light with a wavelength of 596 nm decays to 50% at 800 μm along the main emission axis,¹⁰ which in our case is radially from the sidelight fiber. If the response from the back emission probe is on par with that of the surrounding fibers, it means that it is feasible to put fibers behind laminar probes to stimulate brain tissue."*

3. L In Fig. 2H, optogenetic inhibition effects on neural responses were similar for two different illuminations: one by the BE probe and the other by the fiber matrix. If there are no distinguishable responses, then what is the utility of implanting multiple fibers? A simple flood exposure from a single fiber would do the same effect.

The reviewer is right in that the BE-probe itself generates an illumination and stimulation effect that is comparable to four surrounding fibers. This shows that, at least in this brain area, it is feasible to mount a fiber behind laminar probe (see answer to major point 2 above). In all previous publications the fibers have been placed on the front of the laminar probe but this limits the evenness of the illumination and hampers the recording quality. As we show with a new set of experiments it is even possible to make a depth resolved illumination using multiple staggered fibers behind the laminar probe (please see below).

We confirm this approach in the main text (page 5, lines 203-205):

*"The inhibition was similar for the BE probe and fiber matrix emission (Fig. 2H), which confirms similar PVR amplitudes for both conditions (S**Fig. 5F**). This in turn confirms the validity of placing optical fibers behind laminar probes."*

4. To fully demonstrate the utility of the proposed multichannel optogenetic systems, unique and compelling neuroscience experiments should be presented that could not be achieved by conventional or other state-of-the-art optoelectrodes.

We thank the reviewer for giving us the "push" to show what is possible with this approach. With our new experiments we show that it is possible to simultaneous laminar recordings and multifiber stimulations, three dimensional optogenetic stimulation, causal connectivity inference, behavioral responses to optogenetic stimulation, all being done, in the same freely moving mouse (pages 5-6, lines 213-246).

"Next, we took advantage of the thin optical fibers to allow optogenetic stimulation to occur at different depths in brain tissue. To this end, we bundled 10 fibers of different lengths, resulting in an axial stimulation resolution of 500 μm and a total range of 5 mm (a 250- μm resolution and 2.5-mm range were used for the anterior implantations) (Fig. 3A). A fiber-bundle base diameter of less than 80 μm was achieved by combining fibers with diameters of 11.8 and 30 μm . The 11.8- and 30- μm -diameter fibers were polished to lengths of approximately 500 and 2000 μm , respectively. The 11.8- μm fiber was polished using cooling liquid (milk or water) to avoid heat-related bending of the fibers."

Approximately 70 to 200 fibers were polished simultaneously (**Fig. 3B** and **SFigs. 7A–D**), and approximately 50% of those fibers had an appropriate light distribution (**Fig. 3A**). The fibers on the unpolished side were arranged next to each other with a fiber center-to-center distance of approximately 80 μm (**SFigs. 7E–F**). We implanted six such depth probes bilaterally in a mouse brain (**Fig. 3C**). The connectors were grouped and attached to the mouse skull such that they could be scanned and individual fibers could be identified using a fiber bundle (**Figs. 3D–E**). This fiber bundle allowed us to assemble the connector during implantation and therefore eliminated the constraints inherent in a fixed connector (**Fig. 3F**). The scanned image plane was thus transmitted to the animal connector, and the individual fibers were detected using a photodiode that was submerged in dental cement (**Fig. 3G**). The fiber bundle allowed only minimal light leakage into neighboring fibers because the attenuation between the fibers was close ($\sim 2.5\%$) to that predicted (1%) by the patch cord with an integrated photodiode (**SFig. 7F**).

The fibers were implanted in the primary motor cortex, striatum, and thalamus of two mice. These areas had received virus injections with a channelrhodopsin construct three weeks earlier. The fiber bundle that ended in the left striatum was attached to the backside of a 32-channel silicon probe (similarly to the BE probe described above). With this BE probe, sorted units at different depths could be differentially activated (**Fig. 3H**). The 30- μm fiber was more effective than the 11.8- μm fiber in generating neuronal spiking (**Fig. 3I**). To achieve a behavioral effect with the 11.8- μm fiber, the effect of multiple fibers had to be pooled (**Fig. 3J**). In contrast, a single 30- μm fiber generated reliable behavioral responses, thereby allowing a within-probe comparison of deep versus superficial stimulation (**Fig. 3K**). Finally, we tested whether the multi-areal fiber approach could be combined with extracellular recordings to probe neural connectivity. For fibers ipsilateral of the BE probe, the latency of the neural responses was similar to that generated by the fibers of the BE probe (**Fig 3L**). In contrast, for fibers contralateral to the BE probe, there was a consistent latency shift of approximately 3 to 4 ms. This is consistent with inter-hemispheric connectivity latencies. Thus, this approach shows that it is possible to conduct simultaneous laminar recordings and multifiber stimulations, 3D optogenetic stimulation, connectivity inference, and behavioral quantification in a freely moving animal.“

“Figure 3. Three-dimensional optogenetic stimulation and laminar recordings in freely moving mice. A: Fiber bundle probe used in the experiments for light emission at different depths in the brain tissue.

Scale bar: 1 mm. Schematic side view (left panel); sequentially illuminated fibers lead to defined depth illumination (right panel). B: Multifiber polishing holder and approximately 70 simultaneously polished fibers illuminated with green light for visualization. Scale bar: 4 mm. C: Bilateral ChR2 injection locations, fiber bundle implantation locations, and BE probe implantation illustrated in three exemplary coronal sections. D: Optical set-up for flexible three-dimensional optogenetic stimulation and laminar recordings in a freely moving mouse. A fiber bundle transmits the galvo scanning position to the animal. This eliminates the need to have a tailored patch cord for every individual experiment. Furthermore, because this allows the fiber matrix connector to be built incrementally during the implantation, it allows each fiber probe to be implanted independently, without being constrained by the connector. The fiber bundle (Schott imaging fiber bundle; see methods) was 2 m long and attenuated the light by 75%. E: Implantation schematic. The locations of the fibers in the fiber matrix connector were detected using an implanted photodetector. The fibers emit stray light that can be detected by the photo diode in the transparent cement. This photo diode signal was sent via the commutator to the recording system. F: Connector behind an aligning glass plate during implantation. To make the fibers visible, a white light source was directed toward the mouse skull. The deep 30- μm fibers (Fiber 1) exhibited a red emission with a low intensity (leftmost fiber in each row). The superficial 30- μm fibers (Fiber 6) exhibited a yellowish emission with a higher intensity (middle fiber in each row). The 11.8- μm fibers absorbed a higher percentage of the white light above the brain and therefore emitted white light (scale bar: 500 μm). Some fibers were not covered by the fiber bundle because the connector was aligned by manual adjustments of the original silicon plate. For the fiber bundle, the silicon plate is not necessary, and the connector on the animal side can have hexagonal constraining walls (3D printed or CNC machined) to ensure alignment with the hexagonal fiber bundle. G: Readout of the implanted photodiode during X/Y galvo scanning (scale bar: 500 μm). Voltage is linearly mapped to the jet pseudocolor scale in MATLAB. H: Depth-resolved activation of sorted units (spike shapes are shown in the left column). For each unit, the firing rate was calculated 3–8 ms after light onset (the gray shaded area in the middle column), and the background firing rate preceding light onset was subtracted to quantify the light-evoked rate modulation for different units, depths, and fibers (right column). Fiber 7 was nonfunctional and was removed for clarity. The emitted intensities from the 11.8- and 30- μm fibers were estimated before implantation to be in the ranges of 0.5–1 and 1.5–3 mW, respectively. I: Units with significant activations for 11.8- and 30- μm fibers from two animals. The number of units that responded to the light is denoted by n . The total numbers of units were 12 for animal 960 and 25 for animal 959. J: Optogenetic effect on behavior measured in body rotations. There was a significant difference between left and right rotations depending on whether the 11.8- μm fibers in the left or right hemisphere were stimulated. In both hemispheres, all two, three, four, and five fibers were stimulated in nine trials each, resulting in 36 trials in total for each hemisphere. The amount of body turning was estimated through visual inspection across four cameras. K: Behavioral effect for depth-resolved stimulation in cortex (Ctx) versus striatum (Str) for the 30- μm fibers in the right hemisphere. The behavior was estimated according to panel (J). The number of trials is denoted by n . L: Cortical recordings during the stimulation of six different superficial 30- μm fibers (stimulated fibers are indicated by a blue circle) organized bilaterally at three different positions along the anterior (A)–posterior axis (P). To rule out that longer latencies for the stimulation on the contralateral hemisphere (right column) were due to light leakage to the recorded hemisphere, we calculated the peak latency (vertical dashed blue line) for the minimum light intensity in the recorded hemisphere that generated a response (blue response curve in the middle inset)."

Minor:

1. In Fig. 1A, what is f5 for? Also, indicate where the fiber matrix connector (Fig. 1D,E) is located in Fig. 1A.

We are grateful that the reviewer found this overlooked item. This fiber was used as a dummy fiber at which our initial constant laser source was focused when none of the four fibers were addressed. In this manuscript we used an acoustical optical modulator (AOM) to reduce the light between stimulations. We have now removed this fiber item from Fig 1A.

Also, we have now added an illustration showing where the fiber matrix connector is located relative to the stimulation setup. Fig 1D.

D

2. In Fig. 1D, more detailed explanation is needed how four fibers can be coupled from one through hole.

We have now added this description to the figure legend of panel 1E-F (page 8, lines 322-324):

Panel E:

Panel F:

“E: Coupling from 60- μm fiber on the laser side to two or four 30- μm fibers on the animal side. F: Topological relation between implanted fibers and fiber matrix connector. The through-hole was 75 μm in diameter, which allowed us to insert four 30- μm fibers.”

3. In Fig. 1E, more explanation is needed to clarify why multiple through-holes are needed and how the fibers are coupled as shown in Fig. 1D.

We have now added this motivation to the figure legend of panel 1E (page 8, lines 328-332):

“Multiple through-holes, with a systematic difference in hole diameters, were used to verify process tolerances (see methods: “Producing the fiber connector using MEMS technologies”). Accordingly, across four wafers, the 30- μm fiber fitted into the 31- μm hole (but not into the 30- μm hole), and the 65- μm fiber fitted into the 67- μm hole (but not into the 65- μm hole), thereby confirming the predictable accuracy of the manufacturing process.”

The fibers were coupled to larger 60 μm fibers into 2-4 smaller 30 μm fibers. To illustrate the coupling we added the following panel E to Figure 1:

Panel E:

Panel F:

“E: Coupling from 60- μm fiber on the laser side to two or four 30- μm fibers on the animal side. F: Topological relation between implanted fibers and fiber matrix connector. The through-hole was 75 μm in diameter, which allowed us to insert four 30- μm fibers.”

4. In SFig. 2B and C, put a scale bar. BS should be RS in SFig. 2B.

We have now corrected this.

5. In SFig. 3B, the detailed explanation of how to characterize the emission angle is needed.

We have now added this information in the methods section “Polishing side-emitting fibers” (pages 23-24, line 622-628):

“The emission at a certain angle was calculated from the gamma-uncorrected intensity values summed across all of the pixels of a manually selected region of interest delineating the polished region of the fiber at a certain angle to the camera.

$$\text{Emission at angle} = \sum_{xy \in \text{ROI}(\text{angle})} I_{xy}^{\text{angle}}$$

The images, I_{xy} , were captured with a Samsung S5KGM2 sensor and measured in $25^\circ \pm 5^\circ$ increments. The intensity varies according to $I = I^0 \cos(\text{angle})$, where the angle is zero radially from the fiber and 90° axially/along the fiber, consistent with the properties of a Lambertian-like diagram.”

6. Manufacturing of side-emitting fibers: How accurately can the polished line length be controlled in terms of micrometers? How reproducible is the manufacturing of emission angles and polished lengths?

We have now added this information in the methods section “Polishing side-emitting fibers” (page 23, lines 613-621):

“The variation in the length of the side-emitting segment is approximately $\pm 200 \mu\text{m}$, depending on the skill level of the polisher and the time invested in the polishing process (see polishing video). The reproducibility of the polished lengths was similar for single and multifiber polishing. The emission angles in SFig. 3B are reproducible as long as the nail polish is diluted 1:4 with acetone. The accuracy and reproducibility can also be increased using multifiber polishing followed by post-hoc selection of appropriate fibers. For the depth-resolved probes used in the mouse experiments, fibers were polished simultaneously, after which only the best 50% were used for the final probe. The evenness and length of the side-emission segment was determined by a laser source with a 473-nm wavelength coupled to a standard 200- μm fiber.”

7. In SFig. 5B, it is not clear what the family curves are. Need more explanation.

We thank the reviewer for pointing this out. The necessary information has now been added to the figure legend of SFig 5B (page 18, lines 481-484):

“B: Assessment of BE probe functionality using PVR to ensure that the fibers are intact. Nine of ten testable fibers gave a photovoltaic response (with the remaining four BE probes, there was a problem with the electrode ZIF-clip; see Table 1). The lower three curves represent the PVRs for the IMTEK probes, and the upper seven curves represent the PVRs for the ATLAS probes.”

8. Photo voltaic response of the silicon probe is so severe that it can be used to profile the spatial resolution of optical illumination from the side-emitting fibers. But it would be non-ideal for opto-electrophysiology. How does this affect temporal resolution and accuracy of spike sorting and local field potential?

The photovoltaic response strongly depended on the type of electrode. The ATLAS probe had an order of magnitudes larger response than the IMTEK probe (See SFig. 5B). As a result, for the IMTEK probe spikes could be detected reliably (Fig 2D). It should also be noted that we used sharp light onset according to a step function with a 0.1ms slope. A ramp over the course of one millisecond has been suggested to drastically decrease the PVR. But even in this case it has been suggested that one need artifact removal for being able to detect spikes (see <https://www.ucl.ac.uk/neuropixels/training/2021-neuropixels-course>, 3.7 - Combining Neuropixels with optogenetics - Maxime Beau (UCL)). We regret that we did not implement light ramping in our stimulation in this manuscript. Despite this we could detect spikes with a high fidelity in both the mouse and in the rat. Finally, regarding the local field potential, it should be noted that most probes including Neuronexus and Neuropixel probes all shows photovoltaic responses. Therefore, for local field potential for which it is nontrivial to remove artifacts the only feasible option seems to be single-channel glass coated electrodes.

We have now added this information to the methods section “PVR management procedure” (page 31, lines 965-964):

“The photovoltaic response strongly depended on the type of electrode. The ATLAS probe responses are an order of magnitude larger than those of the IMTEK probe (see SFig. 5B). As a result, for the IMTEK probe, spikes could be detected reliably (Fig. 2D). It should also be noted that we used sharp light onset according to a step function with a 0.1-ms slope. A ramp over the course of 1 ms has been

suggested to drastically decrease the PVR. However, even in this case, it has been suggested that artifact removal is necessary for detection of spikes (see <https://www.ucl.ac.uk/neuropixels/training/2021-neuropixels-course>, 3.7 - Combining Neuropixels with optogenetics - Maxime Beau (UCL)). Fortunately, we were able to detect action potentials with high fidelity in both mice and rats without implementing light ramps.”

9. In SFig. 5F, the BE (back emission) probe gives illumination of light not only behind the electrodes but also in front of electrodes. How do you distinguish which neurons are optically stimulated? What is the spatial resolution of your optogenetic stimulation? In most optoelectrodes, illumination is confined in front of the probe and sometimes illumination intensity is controlled to stimulate a small volume at the distal end of illumination sites.

With the ultrafast inhibition documented in this manuscript we can accurately distinguish between directly and indirectly inhibited neurons. This is because the fastest synaptic connectivity between neurons is larger than 1 ms. Thus, any inhibition effect that occurs faster than 1 ms cannot be due to a synaptic connectivity but has to be directly induced by the optogenetic inhibition.

This we added into the result section (page 5, lines 197-199):

“For example, an indirect effect can be ruled out for the unit shown in Figure 2D because the inhibition was faster than the synaptic conduction between neighboring neurons, i.e., approximately 1 ms.^{23,24}

To selectively stimulate a certain set of neurons with single photon stimulation is just barely possible with a high resolution spatial light modulator (100x100 or more independent phase modulators). Thus, it is not feasible to selectively stimulate certain set of neurons with a single fiber. Even if the brain tissue had zero scattering and we had 10,000 phase modulated micro LEDs on the laminar probe, it would be impossible to claim that only a certain neuron was stimulated since we don't record from all neurons with the laminar probe.

For this reason here we don't aim to stimulate certain sets of neurons. Rather we have taken the approach that is feasible for stimulating small and large population of neurons non-selectively around the fiber. In the new experiment we have shown that it is possible to record optically activated single units at different depths along the electrode shank. In those experiments we can stimulate population of neurons with a resolution of around 500 micrometres. Additional definition of neurons could originate by genetical approaches, e.g. confined opsin expression in specific neuronal subtypes.

We have added this to the discussion (pages 6-7, lines 258-274):

“The spatial resolution in our approach is achieved by placing the fibers in any desired XYZ position around the electrode. Although our fibers have a small diameter of less than 12 μm (core diameter 8 μm) and individual fibers can be separated by this distance along or orthogonally to the shank, the spatial resolution will nevertheless be drastically limited by light scattering in brain tissue. Although micro-LEDs have the same light-scattering limitation, they can be integrated between electrode sites rather than at the edge of the shank. However, this increased spatial resolution comes with the disadvantages that micro-LEDs are limited to one wavelength per LED and that they cause electromagnetic interference artifacts via their stimulation current, thus limiting the temporal resolution. Even though electromagnetic artifacts can be made smaller than the spike detection threshold,¹⁷ they will nevertheless interfere with the spike shape and therefore with spike sorting. Finally, irrespective of how such a selective optical stimulation is achieved, the stimulation specificity cannot be completely verified with extracellular recordings because of their limited spike detection

horizon. Thus, unless recordings can be obtained from all neurons that are affected by the light,²⁹ one must rely on imaging methods for which the plane of focus can be changed to verify the stimulation specificity.³⁰ We therefore have combined laminar recordings with fiber stimulation, because fibers allow great flexibility in multi-areal stimulation, at a scale of hundreds of micrometers to several millimeters, without causing electromagnetic artifacts.”

10. In Fig. 2H, it illustrates both BE probe and fiber matrix have concentric illumination profile. Is it true? Have you characterized the illumination profile around the side-emitting fiber in 360 degrees? The purpose of the side emission is to provide uniform illumination around the fiber in all directions? If the side-emitting fiber has directionality, then how do you align it when you assemble the fibers to give the maximum illumination toward the silicon probe?

We thank the reviewer for bringing up this essential point. We have now quantified the emission intensity in steps of 45° steps around the sidelight fiber in SFig 3C-D. The lower part of panel 3C and D shows that the coating smoothens emission along the fiber as well as circularly around the fiber.

This is added to supplementary figure 3C and D (page 15 for figure and page 16, lines 452-455 for legend):

“C: Uncoated 2-mm fiber in milk (top). Scale bar: 1 mm. The emitted intensities at eight different angles around the probe axis as measured in air (bottom). D: Coated 2-mm fiber in milk (top). Scale bar: 1 mm. Emitted intensities at eight different angles around the probe axis as measured in air (bottom).”

Reviewer #3 (Remarks to the Author):

In this manuscript, Eriksson et al. demonstrated a fiber probe that combined Lambertian side-emitting optical fibers and silicon probes for simultaneous optogenetic stimulation and neural recording. The demonstrated ability of this probe is potentially useful for neuroscience studies, and this reviewer found the ability to individually address the location of light emission with a x/y galvo scanner particularly innovative. It is worth noting that this capability is compatible with behavioral neuroscience experiments in freely moving animals, facilitated by a motorized commutator and an angular encoder that tracks one of the selected fibers despite the rotation of the ferrule. Despite these advances, this reviewer found it difficult to track down important parameters and detailed information related to the experimental implementation of this technology. Additionally, the authors are encouraged to revise the language to improve readability. The following comments are suggested for the authors to address during the revision:

1. It is unclear how the electrophysiological recordings were collected. The authors claimed that the developed fiber probes can achieve “high quality recordings and ultrafast multichannel optogenetic inhibition in freely moving animals.” (Line 82-83) To support the statement, the authors demonstrate Figure 2 and Supplementary Figure 5. However, the authors failed to provide any details about the data acquisition and analysis methods except for a single sentence on Page 4, Line 150-152, “We recorded electrophysiological signals and applied yellow laser light in freely moving rats in sessions of 60 to 120 minutes over the course of 10 days.” For example, are the original recorded traces filtered, and if so, what cutoff frequency and type of filters did the authors apply to the data? What thresholding conditions were applied for sorting the single-unit spikes and how are the spikes clustered? How many neurons can be recorded by the silicon probe per channel, and are all neurons subject to optogenetic inhibition? Fig. 2B shows the response of “units” under illumination, while it remains unclear whether these units are from the same brain region and the same channel of the silicon probe or not. In that sense, the average “event rate” shown at the end of Fig. 2B is poorly defined since neurons of different types usually have vastly different baseline firing rates, ranging from <0.1 Hz to >100 Hz. Therefore, it makes more sense to summarize the results based on each identified single unit instead of showing an average firing rate (e.g., different neurons may also show distinct behavior to illumination, depending on the efficiency of viral transduction and the selectivity of transducing different neurons types based on the promoter type).

We thank the reviewer for bringing up this important point. In reaction to the comment, we added an entire section about “Data acquisition and spike sorting using KiloSort” in the methods (page 29, lines 865-877).

“Data acquisition and spike sorting using KiloSort

Extracellular signals were bandpass-filtered, amplified, and digitized using INTAN (Intan Technologies, Los Angeles, California) two-head stages (RHD2132) that were integrated into zero-insertion-force clips (ZD32, Tucker Davis Technologies) for the bilateral BE probe rat experiments and one-head stage (RHD2132) that was integrated into zero-insertion-force clips (ZD32, Tucker Davis Technologies) for the unilateral BE probe mouse experiments. To maximize the animals’ comfort, we suspended the ultrathin INTAN cable by an ultralight spring with a 1.5-mm diameter.

The extracellular recordings were filtered with a 7.5-kHz low-pass cutoff (third-order Butterworth) and a 0.1-Hz high-pass cutoff (first-order Butterworth) and sampled at 30 kHz, after which the signal was digitally high-pass filtered with a 1.0-Hz cutoff (first-order Butterworth). The PVR was removed off-line (see “PVR management procedure” below). Spike sorting was done using KiloSort with default parameters. Units with a clear PVR response 1 ms after the light onset were not used for further analysis.”

To quantify the inhibition effect in relation to the baseline firing rate we added supplementary figure 6 (page 18, lines 502-506).

“Supplementary Figure 6. Relation between baseline firing rate and firing rate during optogenetic inhibition. In areas with strong expression (yellow dots), a large proportion of neurons were inhibited (35% of the neurons were more than 90% inhibited compared to baseline activity). In areas with low expression, the inhibition effect was visibly reduced (black dots).”

In this figure we also separated units that were recorded in a region that had low EYFP expression and units that were in a region with high EYFP expression. In areas with strong expression (yellow dots), a large fraction of neurons were inhibited (35% of the neurons were more than 90% inhibited compared to baseline activity). In areas with low expression the inhibition effect was visibly reduced (black dots). Finally, we kindly refer the reviewer to supplementary figure 5I for number of units per electrode.

Further, according to the reviewer’s suggestion we send the manuscript for professional proof reading and language revision to Cambridge ProofReading LCC (please see certificate).

2. It is unclear whether the bundle of fiber probes can avoid blood vessels damages. The authors claimed that the bundle of fiber probes “allow any combination of areas to be addressed, as well as individualized designs for each animal to avoid blood vessels and to target areas” The authors may include additional immunohistochemistry assays to prove the claim. Another missing information is the distance between the Si probe and the optical fibers, and a scale bar is needed for the schematics in Fig. 2A, those in the left column of Fig. 2H, Supplementary Fig. 4A, and Supplementary Fig. 5A. This information is important since single-unit recordings in the bottom panel of Fig. 2H show significant decreased firing even when the illuminated areas are far away from the silicon probe. The authors are encouraged to refer to “Spatiotemporal constraints on optogenetic inactivation in cortical circuits” *elife* 2019 for the radius of inactivation and compare it to their measurements in Fig. 2H.

The reviewer’s comment made us realize that our statement about avoiding blood vessels can be interpreted differentially from what we intended. We only referred to the avoidance of superficial blood vessels which can be identified on the brains surface by visual inspection during the stereotactic implantation. This is because the fibers are very thin and multiple fibers can be placed independently of each other and independently of the connector such that blood vessels on the surface of the brain can be avoided. Of course, the approach does not guarantee that deep blood vessels are avoided and we assume that this was the point of confusion. We now clarified this statement in the main text (page 7, lines 302-304):

“This simplifies not only the targeting of multiple areas but also the targeting of areas defined by fluorescence while avoiding superficial blood vessels.”

Regarding the range of the inhibition we thank the reviewer for this comment and the suggestion to relate our data to that of the elegant bleaching assay in “Spatiotemporal constraints on optogenetic inactivation in cortical circuits”. According to this study, the intensity of yellow light decreases to around 50% after roughly 800 μm along the major emission direction. In the elife paper, the major emission direction is axially while in our sidelight fibers the major emission direction is radially. Thus the 50% decrease is within the range of our surrounding fibers which are at a distance of roughly 700 μm from the BE-probe. We now discuss this differences and cite the Elife paper (page 4, lines 151-155):

“To test the validity of this back emission (BE) probe, it was surrounded by four additional fibers (hereinafter referred to as the “fiber matrix”) at a distance of 700 μm (SFig. 4B). Light with a wavelength of 596 nm decays to 50% at 800 μm along the main emission axis,¹⁰ which in our case is radially from the sidelight fiber.”

Further, we have now added a scale bar to the schematics in Fig. 2A.

3. This reviewer cannot find the exact stereotaxic coordinates in the “Animal surgery and BE-probe implantation” method section except for a (very wide) range of locations. It is known that the neuron density and types vary significantly from different brain regions and the authors need to use a consistent brain region for rigorous scientific demonstration of their approaches.

This information has now been added in the figure panel 2A.

4. In comparison to previously reported tapered fibers, the authors state that “Although the elegant tapered fibers produce a more even illumination along the electrode shank, the light is typically strongest towards the fiber tip.” However, the author did not include a quantitative comparison of the intensity profiles between their optical fibers and the tapered fibers. Please perform quantitative analysis of the intensity profiles for images shown in Fig. 1I and Supplementary Fig. 3 to substantiate this claim. Since the authors now have demonstrated individually addressed fiber output based on raster scan, can they adjust the input angle and location of illumination during the programmed scan to achieve more uniform intensity distribution without using a diffusing layer?

We have now added this quantification in supplementary figure 3H-J:

“H: Quantification of the intensity profile along a Lambda fiber in cortex at different distances from the probe. Each curve was calculated according to the distance at which the maximal intensity (along the probe axis) corresponded to a certain percentage (in increments of 10%) of the maximal intensity (at the probe). The underlying intensity map (bottom) was retrieved from Figure 2C in Pisanello et al. (2017) (scale bar: 500 μm). I: The same as in H but for a coated 30- μm fiber in milk (scale bar: 1 mm). J: The same as in H but for an uncoated 30- μm fiber in milk (scale bar: 1 mm).”

We now also refer to supplementary figure 3H-J in the main text (page 4, lines 147-149):

“The ability to control how much light is emitted at each position along the fiber independently (through manual polishing), together with the diffusing layer, results in more even emission along the fiber relative to the Lambda probe (S Figs. 3H–J).”

5. It is important to demonstrate whether the fiber probe is small and light enough to be used in freely moving animals. The authors may include additional behavioral assays to convince potential readers.

We thank the reviewer for giving us the “push” to show what is possible with this approach. With our new experiments we show that it is possible to simultaneous laminar recordings and multifiber stimulations, three dimensional optogenetic stimulation, causal connectivity inference, behavioral responses to optogenetic stimulation, all being done, in the same freely moving mouse (pages 5-6, lines 213-246).

“Next, we took advantage of the thin optical fibers to allow optogenetic stimulation to occur at different depths in brain tissue. To this end, we bundled 10 fibers of different lengths, resulting in an axial stimulation resolution of 500 μm and a total range of 5 mm (a 250- μm resolution and 2.5-mm range were used for the anterior implantations) (Fig. 3A). A fiber-bundle base diameter of less than 80 μm was achieved by combining fibers with diameters of 11.8 and 30 μm . The 11.8- and 30- μm -diameter fibers were polished to lengths of approximately 500 and 2000 μm , respectively. The 11.8- μm fiber was polished using cooling liquid (milk or water) to avoid heat-related bending of the fibers. Approximately 70 to 200 fibers were polished simultaneously (Fig. 3B and S Figs. 7A–D), and approximately 50% of those fibers had an appropriate light distribution (Fig. 3A). The fibers on the unpolished side were arranged next to each other with a fiber center-to-center distance of approximately 80 μm (S Figs. 7E–F). We implanted six such depth probes bilaterally in a mouse brain (Fig. 3C). The connectors were grouped and attached to the mouse skull such that they could be scanned and individual fibers could be identified using a fiber bundle (Figs. 3D–E). This fiber bundle allowed us to assemble the connector during implantation and therefore eliminated the constraints inherent in a fixed connector (Fig. 3F). The scanned image plane was thus transmitted to the animal connector, and the individual fibers were detected using a photodiode that was submerged in dental cement (Fig. 3G). The fiber bundle allowed only minimal light leakage into neighboring fibers because the attenuation between the fibers was close ($\sim 2.5\%$) to that predicted (1%) by the patch cord with an integrated photodiode (S Fig. 7F).

The fibers were implanted in the primary motor cortex, striatum, and thalamus of two mice. These areas had received virus injections with a channelrhodopsin construct three weeks earlier. The fiber bundle that ended in the left striatum was attached to the backside of a 32-channel silicon probe (similarly to the BE probe described above). With this BE probe, sorted units at different depths could be differentially activated (Fig. 3H). The 30- μm fiber was more effective than the 11.8- μm fiber in generating neuronal spiking (Fig. 3I). To achieve a behavioral effect with the 11.8- μm fiber, the effect of multiple fibers had to be pooled (Fig. 3J). In contrast, a single 30- μm fiber generated reliable behavioral responses, thereby allowing a within-probe comparison of deep versus superficial stimulation (Fig. 3K). Finally, we tested whether the multi-areal fiber approach could be combined with extracellular recordings to probe neural connectivity. For fibers ipsilateral of the BE probe, the latency of the neural responses was similar to that generated by the fibers of the BE probe (Fig 3L). In contrast, for fibers contralateral to the BE probe, there was a consistent latency shift of approximately 3 to 4 ms. This is consistent with inter-hemispheric connectivity latencies. Thus, this approach shows that it is possible to conduct simultaneous laminar recordings and multifiber

stimulations, 3D optogenetic stimulation, connectivity inference, and behavioral quantification in a freely moving animal."

“Figure 3. Three-dimensional optogenetic stimulation and laminar recordings in freely moving mice. A: Fiber bundle probe used in the experiments for light emission at different depths in the brain tissue. Scale bar: 1 mm. Schematic side view (left panel); sequentially illuminated fibers lead to defined depth illumination (right panel). B: Multifiber polishing holder and approximately 70 simultaneously polished fibers illuminated with green light for visualization. Scale bar: 4 mm. C: Bilateral ChR2 injection locations, fiber bundle implantation locations, and BE probe implantation illustrated in three exemplary coronal sections. D: Optical set-up for flexible three-dimensional optogenetic stimulation and laminar recordings in a freely moving mouse. A fiber bundle transmits the galvo scanning position to the animal. This eliminates the need to have a tailored patch cord for every individual experiment. Furthermore, because this allows the fiber matrix connector to be built incrementally during the implantation, it allows each fiber probe to be implanted independently, without being constrained by the connector. The fiber bundle (Schott imaging fiber bundle; see methods) was 2 m long and attenuated the light by 75%. E: Implantation schematic. The locations of the fibers in the fiber matrix connector were detected using an implanted photodetector. The fibers emit stray light that can be detected by the photo diode in the transparent cement. This photo diode signal was sent via the commutator to the recording system. F: Connector behind an aligning glass plate during implantation. To make the fibers visible, a white light source was directed toward the mouse skull. The deep 30- μm fibers (Fiber 1) exhibited a red emission with a low intensity (leftmost fiber in each row). The superficial 30- μm fibers (Fiber 6) exhibited a yellowish emission with a higher intensity (middle fiber in each row). The 11.8- μm fibers absorbed a higher percentage of the white light above the brain and therefore emitted white light (scale bar: 500 μm). Some fibers were not covered by the fiber bundle because the connector was aligned by manual adjustments of the original silicon plate. For the fiber bundle, the silicon plate is not necessary, and the connector on the animal side can have hexagonal constraining walls (3D printed or CNC machined) to ensure alignment with the hexagonal fiber bundle. G: Readout of the implanted photodiode during X/Y galvo scanning (scale bar: 500 μm). Voltage is linearly mapped to the jet pseudocolor scale in MATLAB. H: Depth-resolved activation of sorted units (spike shapes are shown in the left column). For each unit, the firing rate was calculated 3–8 ms after light onset (the gray shaded area in the middle column), and the background firing rate preceding light onset was subtracted to quantify the light-evoked rate modulation for different units, depths, and fibers (right column). Fiber 7 was nonfunctional and was removed for clarity. The emitted intensities from the 11.8- and 30- μm fibers were estimated before implantation to be in the ranges of 0.5–1 and 1.5–3 mW, respectively. I: Units with significant activations for 11.8- and 30- μm fibers from two animals. The number of units that responded to the light is denoted by n. The total numbers of units were 12 for animal 960 and 25 for animal 959. J: Optogenetic effect on behavior measured in body rotations. There was a significant difference between left and right rotations depending on whether the 11.8- μm fibers in the left or right hemisphere were stimulated. In both hemispheres, all two, three, four, and five fibers were stimulated in nine trials each, resulting in 36 trials in total for each hemisphere. The amount of body turning was estimated through visual inspection across four cameras. K: Behavioral effect for depth-resolved stimulation in cortex (Ctx) versus striatum (Str) for the 30- μm fibers in the right hemisphere. The behavior was estimated according to panel (J). The number of trials is denoted by n. L: Cortical recordings during the stimulation of six different superficial 30- μm fibers (stimulated fibers are indicated by a blue circle) organized bilaterally at three different positions along the anterior (A)–posterior axis (P). To rule out that longer latencies for the stimulation on the contralateral hemisphere (right column) were due to light leakage to the recorded hemisphere, we calculated the peak latency (vertical dashed blue line) for the minimum light intensity in the recorded hemisphere that generated a response (blue response curve in the middle inset).”

6. Typos

a. Page 3, Line 91-92: “as well as well as” should be changed to “as well as”

This typo has been corrected.

b. Page 4, Line 168-169: The authors wrote “SFig. 5K and K.”

This typo has been corrected to **SFig. 5J and K.**

c. It is recommended to change “Photo Voltaic” to “Photovoltaic”

This has been corrected.

REVIEWERS' COMMENTS

Reviewer #1 (Remarks to the Author):

I thank the authors for the constructive approach. The revised version of the manuscript answers to my questions and concerns, and has now more clear and specific claims.

I'm therefore supporting for publication in this present form.

Reviewer #2 (Remarks to the Author):

Authors well addressed all the questions and comments that the reviewers raised. One minor thing is that the y-axis intensity scale 1% should be checked if it is correct in Supplementary Figure 3, C and D.

Reviewer #3 (Remarks to the Author):

The authors have performed an impressive array of additional experiments to address my comments as well as those from other reviewers. I found their responses sufficient and the revised manuscript suitable for publication in Nature Communications. I hereby give my full support for the acceptance of this paper.

Reviewer #1 (Remarks to the Author):

I thank the authors for the constructive approach. The revised version of the manuscript answers to my questions and concerns, and has now more clear and specific claims.

I'm therefore supporting for publication in this present form.

We thank the reviewer for the rigorous methodological comments and for making the manuscript more stringent!

Reviewer #2 (Remarks to the Author):

Authors well addressed all the questions and comments that the reviewers raised. One minor thing is that the y-axis intensity scale 1% should be checked if it is correct in Supplementary Figure 3, C and D.

Indeed the intensity is supposed to be maximally 100% and not 1%. We have now corrected this.

We thank the reviewer for pointing us towards previously missing relevant literature and for pushing us to do better experiments!

Reviewer #3 (Remarks to the Author):

The authors have performed an impressive array of additional experiments to address my comments as well as those from other reviewers. I found their responses sufficient and the revised manuscript suitable for publication in Nature Communications. I hereby give my full support for the acceptance of this paper.

We thank the reviewer for the rigorous methodological comments and for the suggestions to invest in more convincing experiments!